# Decoupling light absorption and carrier transport via heterogeneous doping in Ta$_3$N$_5$ thin film photoanode

Yequan Xiao[1], Zeyu Fan[1], Mamiko Nakabayashi[2], Qiaoqiao Li[3], Liujiang Zhou[3], Qian Wang[4,5], Changli Li[6], Naoya Shibata[2], Kazunari Domen[7,8] & Yanbo Li[1] ✉

The trade-off between light absorption and carrier transport in semiconductor thin film photoelectrodes is a major limiting factor of their solar-to-hydrogen efficiency for photoelectrochemical water splitting. Herein, we develop a heterogeneous doping strategy that combines surface doping with bulk gradient doping to decouple light absorption and carrier transport in a thin film photoelectrode. Taking La and Mg doped Ta$_3$N$_5$ thin film photoanode as an example, enhanced light absorption is achieved by surface La doping through alleviating anisotropic optical absorption, while efficient carrier transport in the bulk is maintained by the gradient band structure induced by gradient Mg doping. Moreover, the homojunction formed between the La-doped layer and the gradient Mg-doped layer further promotes charge separation. As a result, the heterogeneously doped photoanode yields a half-cell solar-to-hydrogen conversion efficiency of 4.07%, which establishes Ta$_3$N$_5$ as a leading performer among visible-light-responsive photoanodes. The heterogeneous doping strategy could be extended to other semiconductor thin film light absorbers to break performance trade-offs by decoupling light absorption and carrier transport.

Efficient light absorption and carrier transport are two essential aspects for achieving high solar-to-hydrogen efficiency in photoelectrochemical (PEC) water splitting devices[1-4]. However, in most of the PEC materials including $\alpha$-Fe$_2$O$_3$ (ref. 5), BiVO$_4$ (ref. 6), Cu$_2$O (ref. 7), and Ta$_3$N$_5$ (ref. 8), it is challenging to achieve efficient light absorption and carrier transport simultaneously. Because the optical absorption depth is significantly larger than the carrier diffusion length in these PEC materials[9,10], improvement on one aspect often comes with trade-offs on the other. Decoupling light absorption and carrier transport is therefore required to avoid performance trade-offs. A common strategy to decouple light absorption and carrier transport is utilizing nanostructured photoelectrodes[7,11-17]. However, the increased junction area in nanostructured photoelectrodes decreases the photon flux received by the semiconductor per unit area, which in theory may reduce the attainable photovoltage of the semiconductor liquid junction[11]. Consequently, the thermodynamic driving force for water splitting is lowered and the photocurrent onset potential of the nanostructured photoelectrode is negatively affected. For instance, although photocurrent density approaching its theoretical value has been achieved with nanostructured Ta$_3$N$_5$ photoanodes[17,18], the onset potential is above 0.6 V versus reversible hydrogen electrode (RHE), which is significantly higher than its theoretical value (<0 V versus

[1]Institute of Fundamental and Frontier Sciences, University of Electronic Science and Technology of China, Chengdu 610054, China. [2]Institute of Engineering Innovation, The University of Tokyo, Tokyo 113-8656, Japan. [3]School of Physics, University of Electronic Science and Technology of China, Chengdu 610054, China. [4]Graduate School of Engineering, Nagoya University, Nagoya 464-8603, Japan. [5]Institute for Advanced Research, Nagoya University, Nagoya 464-8601, Japan. [6]School of Materials, Sun Yat-sen University, Guangzhou 510275, China. [7]Office of University Professors, The University of Tokyo, Tokyo 113-8656, Japan. [8]Research Initiative for Supra-Materials (RISM), Shinshu University, Nagano 380-8553, Japan. ✉e-mail: yanboli@uestc.edu.cn

RHE). As a result, the maximum half-cell solar-to-hydrogen conversion efficiency (HC-STH) achieved by nanostructured Ta$_3$N$_5$ photoanodes is 2.72%, which is significantly lower than its theoretical value (15.9%). Moreover, the thin-film photoelectrodes are likely more durable and less susceptible to corrosion than nanostructured photoelectrodes, and more suitable for scaling up. Nevertheless, an effective strategy to decouple light absorption and carrier transport in thin film Ta$_3$N$_5$ photoanode is desired to further improve its efficiency. Here, we demonstrate a heterogeneous doping strategy that combines surface La doping with bulk gradient Mg doping in a Ta$_3$N$_5$ thin film to decouple light absorption and carrier transport, resulting in a record-high HC-STH of 4.07% for PEC water splitting.

## Results

### Effects of La doping in Ta$_3$N$_5$ thin film
Recent studies revealed that Ta$_3$N$_5$ had significant optical anisotropy along different crystallographic directions[19–22]. Two absorption edges coexist in the Ta$_3$N$_5$ at around 590 nm (~2.1 eV) and 480 nm (~2.6 eV), which are attributed to photon absorption along the *a*-axis and along the *b*- or *c*-axis, respectively. This optical anisotropy leads to less efficient light absorption in the wavelength range of 480–590 nm in Ta$_3$N$_5$ thin films[23,24]. We find that La doping could effectively tune the optical anisotropy of Ta$_3$N$_5$ thin film, resulting in enhanced visible light absorption in the range of 480–590 nm. La-doped Ta$_3$N$_5$ thin films with a La/Ta ratio of 3% and a thickness of 100 nm were prepared on Nb or quartz glass substrates by dual-source electron-beam evaporation of La$_2$O$_3$ and Ta$_2$O$_5$ precursors and followed by nitriding at 1273 K under NH$_3$ flow. Ultraviolet-visible (UV-vis) absorption spectra in Fig. 1a and the corresponding Tauc plots in Supplementary Fig. 1 reveal that there are indeed two absorption edges at around 480 nm (~2.6 eV) and 590 nm (~2.1 eV) in the undoped and La-doped Ta$_3$N$_5$ thin films. However, the light absorption in the range of 480–590 nm is notably enhanced in La-doped Ta$_3$N$_5$ thin film. The conduction band (CB) of Ta$_3$N$_5$ mainly consists of the unoccupied Ta 5*d* orbitals, while the valence band (VB) is mainly composed of N 2*p* orbitals[20]. La has a similar valence electron configuration to that of Ta (La: 5*d*$^1$6*s*$^2$, Ta: 5*d*$^3$6*s*$^2$) and a higher 5*d* orbital energy due to the special 4*f* electron configuration (4*f*$^0$). When Ta is partially substituted by La in Ta$_3$N$_5$, the hybridization of La 5*d* and Ta 5*d* orbitals may lead to more delocalized orbital distribution in the CB, which may contribute to the enhanced light absorption. The first-principles density functional theory (DFT) calculations confirm that both the pristine and La-doped Ta$_3$N$_5$ exhibit two direct band gaps of ~2.1 eV at Γ point and 2.5–2.6 eV at X point (Fig. 1b), in agreement with experimental results. In addition, the DFT calculations reveal that La doping does not generate any intermediate states in the bandgap of Ta$_3$N$_5$. Instead, it leads to increased density of states near the conduction band minimum (CBM) and valence band maximum (VBM) (Supplementary Fig. 2), which accounts for the enhanced light absorption in the range of 480–590 nm.

The UV-vis absorption spectra in Fig. 1a also reveal that the sub-bandgap absorption beyond 600 nm is reduced in La-doped Ta$_3$N$_5$ thin film, indicating the defect density is lowered by La doping. To confirm this, low-temperature photoluminescence (PL) and time-resolved PL (TRPL) were employed to characterize the defect-related emissions and the carrier kinetics in the samples. Under the excitation of a 510-nm laser at 10 K, the defect-related emission at around 800 nm is significantly reduced in La-doped Ta$_3$N$_5$ thin film (Fig. 1c). The PL spectra of the two samples could be deconvolved into three distinctive emission peaks centered at ca. 620 nm (~2.02 eV), 780 nm (~1.59 eV), and 840 nm (~1.48 eV) (Fig. 1d), which are ascribed to defect emissions of residual oxygen impurities substitutionally incorporated on nitrogen sites (O$_N$), nitrogen vacancies (V$_N$) and reduced Ta$^{3+}$ species, respectively. According to our previous study[23,24], O$_N$ defects are shallow donors that improve the conductivity of Ta$_3$N$_5$, while V$_N$ and Ta$^{3+}$ defects are deep traps that act as recombination centers of photocarriers. From Fig. 1d, the PL spectrum of the La-doped Ta$_3$N$_5$ sample

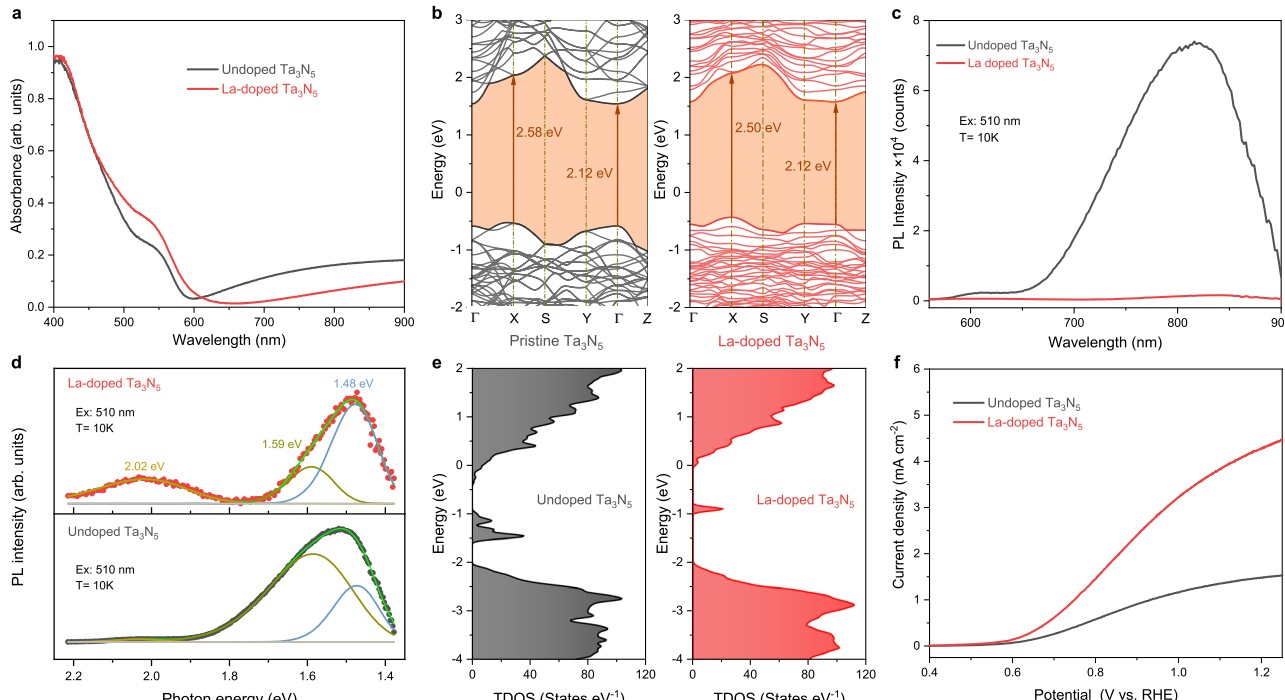

**Fig. 1 | Effect of La doping on the optical and PEC properties of Ta$_3$N$_5$. a** UV-vis absorption spectra of undoped Ta$_3$N$_5$ and La-doped Ta$_3$N$_5$ thin films on quartz substrates. **b** The calculated electron energy bands using MBJ exchange potential for pristine Ta$_3$N$_5$ and La-doped Ta$_3$N$_5$. The brown arrows mark the optical transition gaps. **c** Low-temperature PL spectra of undoped Ta$_3$N$_5$ and La-doped Ta$_3$N$_5$ thin films on quartz substrates measured at 10 K under 510 nm laser excitation. **d** Deconvolution of the PL spectra for the samples. **e** Total density of states (TDOS) calculated by PBE exchange potential for undoped Ta$_3$N$_5$ and La-doped Ta$_3$N$_5$ with intrinsic V$_N$, O$_N$, and Ta$^{3+}$ defects. **f** *J*–*V* curves of undoped Ta$_3$N$_5$ and La-doped Ta$_3$N$_5$ photoanodes with NiCoFe-B$_i$ co-catalyst in 1 M KOH under AM 1.5 G simulated sunlight.

shows an increase in the relative intensity of the $O_N$ defects emission with respect to the deep-level defects ($V_N$ and $Ta^{3+}$) emission, indicating that La doping introduces more $O_N$ donors while suppressing $V_N$ and $Ta^{3+}$ traps in $Ta_3N_5$ film. TRPL decay curves recorded for the near-band-edge emission (Supplementary Fig. 3) show that the average lifetime in La-doped $Ta_3N_5$ thin film is longer than in undoped $Ta_3N_5$ thin film, which is attributed to the reduced defect density in the film. The reduction of defect states by La doping is further supported by DFT calculations. For the undoped $Ta_3N_5$ with intrinsic $O_N$, $V_N$, and $Ta^{3+}$ defects, the intermediate states mainly consist of Ta 5d and N 2p orbitals (Supplementary Fig. 4a). Upon doping La into $Ta_3N_5$ lattice, the intermediate states mostly originate from Ta 5d, N 2p orbitals, and a small amount of La 5d orbital (Supplementary Fig. 4b). The calculated total density of states for undoped $Ta_3N_5$ and La-doped $Ta_3N_5$ are plotted in Fig. 1e. The undoped $Ta_3N_5$ shows a high density of intermediate states, which is consistent with our previous results[23]. In contrast, the density of intermediate states is significantly reduced in La-doped $Ta_3N_5$.

To further evaluate the effect of La doping in $Ta_3N_5$, X-ray photoelectron spectroscopy (XPS) was carried out and the survey spectra are presented in Supplementary Fig. 5. The core-level spectra of La 3d, Ta 4f, O 1s, and N 1s are provided in Fig. 2. The La 3d spectrum (Fig. 2a) of La dopants in $Ta_3N_5$ has well-separated $3d_{5/2}$ and $3d_{3/2}$ spin-orbit components, each of which can be deconvoluted into two sets of

Gaussian peaks. For the $3d_{5/2}$ component, the peaks at 830.8 and 834.7 eV are attributed to $La^{3+}$ in the oxidation state and its satellites peak[25,26], respectively, while the peaks at 837.3 and 840.9 eV correspond to La-N bonds[27,28]. Quantitative XPS analyses show that the La/Ta ratio in the La-doped $Ta_3N_5$ thin film is 2.97%. Figure 2b shows the core-level XPS spectra of Ta 4f region. For undoped $Ta_3N_5$ thin film, there are three doublets (Ta $4f_{7/2}$-Ta $4f_{5/2}$) attributable to the reduced $Ta^{3+}$ species, $Ta_3N_5$ (N-Ta-N) and $TaO_xN_y$ (N-Ta-O)[24]. With La doping, the signal of reduced $Ta^{3+}$ species almost completely disappeared, indicating that La doping can effectively suppress the formation of reduced $Ta^{3+}$ defects, consistent with the PL and DFT results. The atomic ratios of N/Ta and O/Ta in the samples were quantified by combining the Ta $4f_{7/2}$ signal with the N 1s peak (Fig. 2c) and O 1s lattice oxygen peak (Fig. 2d). With La doping, the O/Ta atom ratio in $Ta_3N_5$ thin film increases from 0.27 to 0.32, while the N/Ta atom ratio decreases slightly from 1.59 to 1.58, suggesting that La doping can introduce more $O_N$ donors into $Ta_3N_5$ lattice. For Mg-doped $Ta_3N_5$ film, both nitrogen and oxygen contents are increased, the N/Ta atom ratio is 1.62 and the O/Ta atom ratio is 0.30 (Supplementary Fig. 6), which is consistent with our previous results[24]. The increased density of $O_N$ donors can improve the carrier concentration and promote charge transfer[29–32]. Indeed, a Mott-Schottky analysis (Supplementary Fig. 7) shows that the carrier concentration in the La-doped $Ta_3N_5$ thin film has a ~2.5-fold increase compared to the undoped sample.

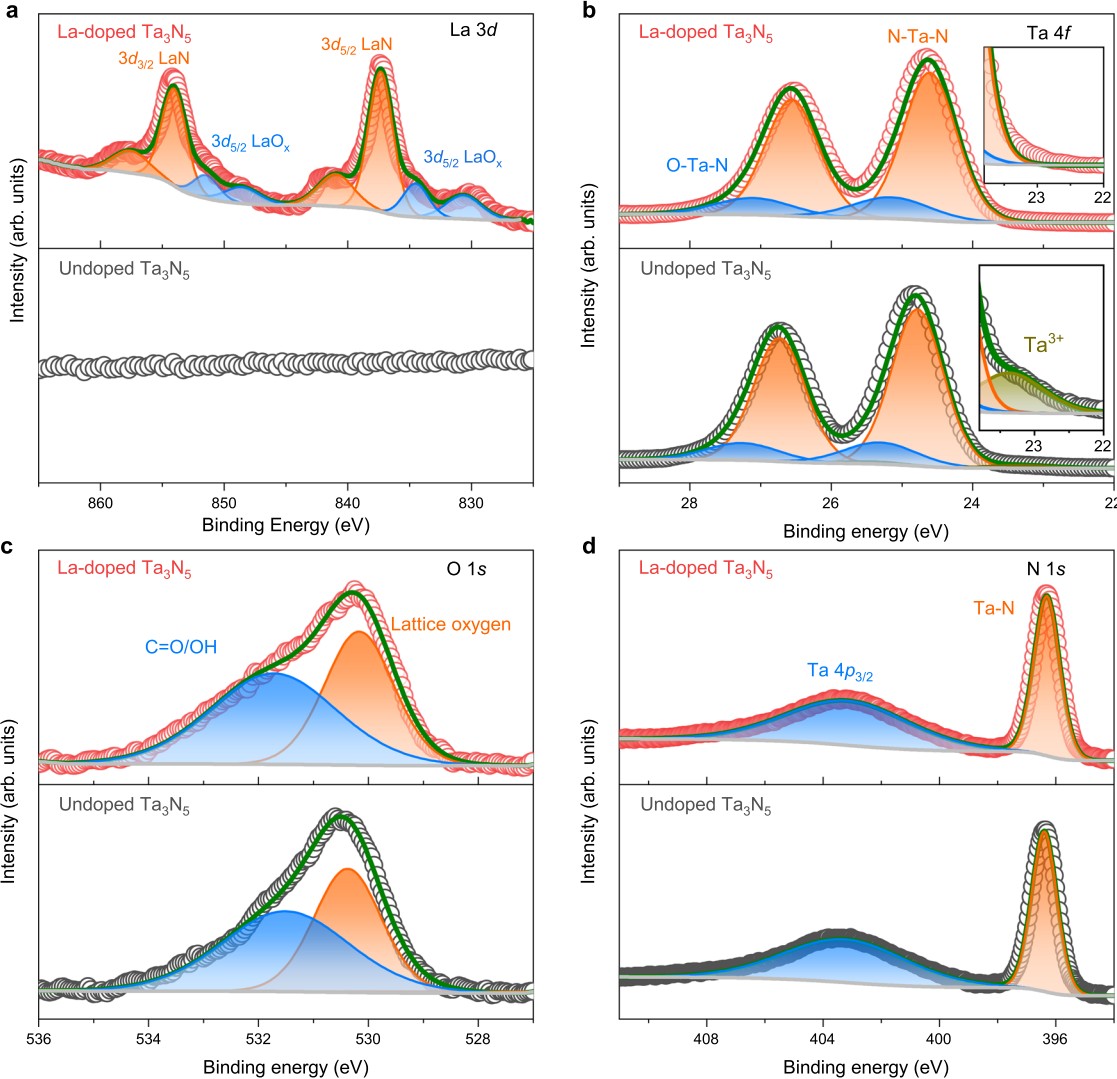

**Fig. 2 | XPS core-level spectra of undoped $Ta_3N_5$ and La-doped $Ta_3N_5$. a** La 3d peaks. **b** Ta 4f peaks. **c** O 1s peaks. **d** N 1s peaks.

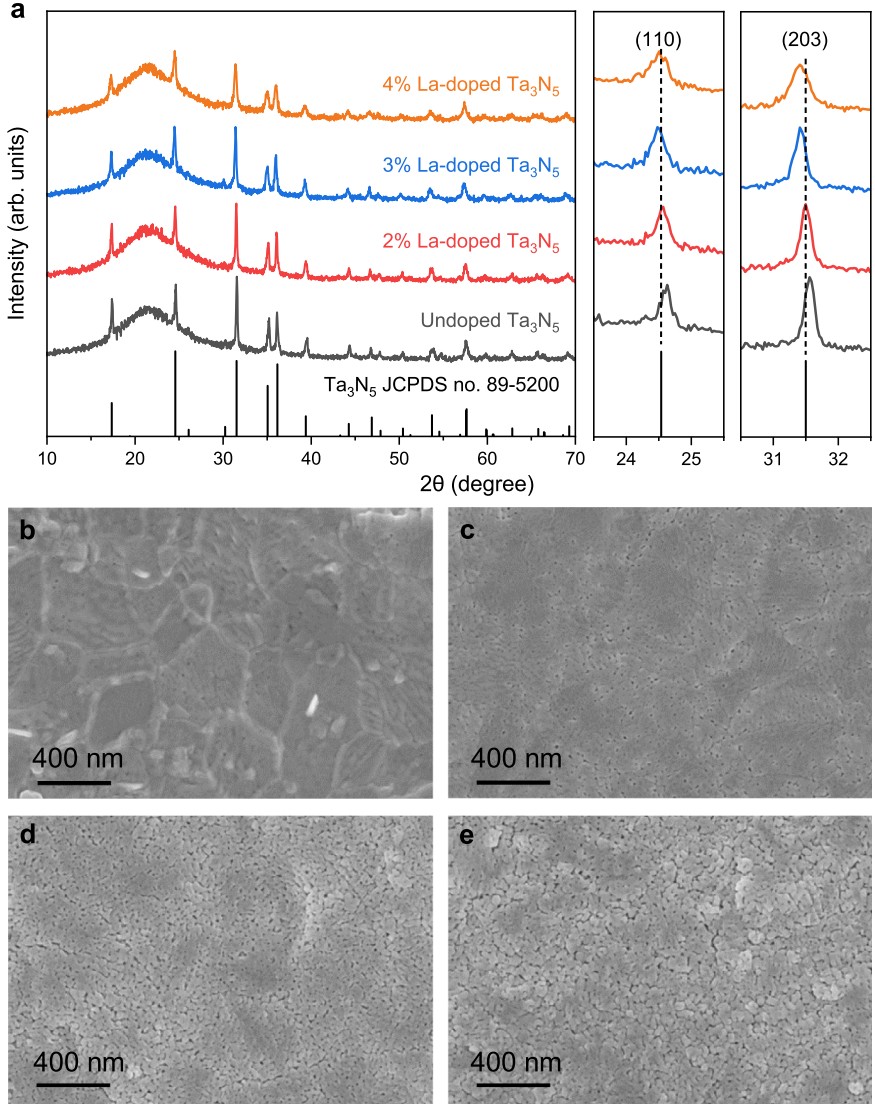

**Fig. 3 | Effect of La doping on the crystallinity of the Ta₃N₅ thin films. a** X-ray diffraction patterns for undoped and La-doped Ta₃N₅ films on quartz substrates. The zoomed-in image on the right side shows the Ta₃N₅ (110) and (203) diffraction peaks. **b**–**e** SEM images for undoped Ta₃N₅ film (**b**), 2% La-doped Ta₃N₅ film (**c**), 3% La-doped Ta₃N₅ film (**d**), and 4% La-doped Ta₃N₅ film (**e**), respectively.

Furthermore, the XPS valence band spectra (Supplementary Fig. 8) reveal that the gap between the Fermi level ($E_F$) and the valence band maximum (VBM) of the La-doped Ta₃N₅ thin film increases by 0.23 eV relative to that of the undoped Ta₃N₅ film. The increase of Fermi level further confirms the increase of carrier concentration in La-doped Ta₃N₅ thin film. A similar enhancing effect of La doping on the carrier concentration of Ta₃N₅ has been reported[33].

Owing to the increased light absorption, reduced deep-level defects, and improved conductivity, photoanode based on La-doped Ta₃N₅ thin film with a thickness of only 100 nm achieves an impressive photocurrent density of 4.40 mA cm⁻² at 1.23 V versus RHE under AM 1.5G simulated sunlight, as compared to 1.51 mA cm⁻² for the undoped sample under the same conditions (Fig. 1f). However, the enhancing effect of La doping on the PEC performance of Ta₃N₅ thin film is found to be highly dependent on the thickness of the films (Supplementary Table 1). For undoped Ta₃N₅ photoanodes, the light absorption and photocarrier utilization are enhanced with increasing film thickness up to 700 nm, resulting in a continuous increase in photocurrent (Supplementary Fig. 9a, c). For La-doped Ta₃N₅ photoanodes, the photocurrent was improved to a lesser extent at a thickness of 300 nm, and even decreased for thicker (500 and 700 nm) films (Supplementary

Fig. 9b, d). The reason for the performance decay in thicker samples is mainly due to decreased grain size of the La-doped Ta₃N₅ thin film, as revealed below.

## Limitations of La doping in Ta₃N₅ photoanode

La-doped Ta₃N₅ thin films (100 nm thick) with different La doping concentrations (2%, 3%, and 4%) were prepared. The actual La/Ta ratios in 2%, 3%, and 4% La-doped Ta₃N₅ thin films were 1.86%, 2.77%, and 4.14%, respectively, as determined by energy dispersive X-ray spectroscopy (EDS) (Supplementary Figs. 10–11). Among these samples, the 3% La-doped Ta₃N₅ photoanode achieved the highest PEC performance (Supplementary Fig. 12). The crystal structure of the thin films on quartz substrate was determined by X-ray diffraction (XRD). Figure 3a shows the XRD patterns of the undoped and La-doped Ta₃N₅ thin films. All samples exhibit an identical crystal structure with orthorhombic anosovite (Ti₃O₅) phase, which demonstrates that La doping does not alter the intrinsic crystal structure of the Ta₃N₅ films. However, the diffraction peaks are slightly shifted to lower angles as the La doping level increases, indicating a lattice expansion induced by La dopants. Considering that six-coordinate La³⁺ have significantly larger ionic radii (103 pm) than six-coordinate Ta⁵⁺ (64 pm), the

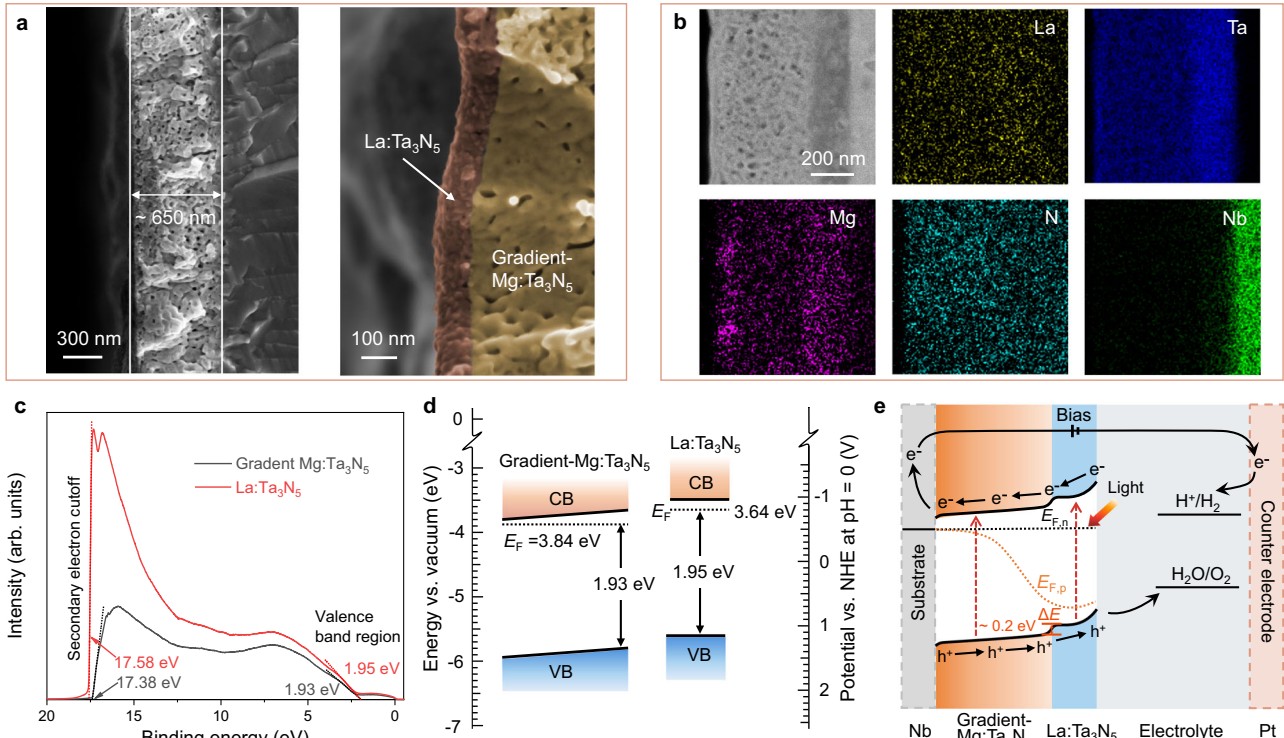

**Fig. 4 | Electron microscopic and band edge energetics characterizations of Nb/gradient-Mg:Ta₃N₅/La:Ta₃N₅.** **a** Cross-sectional SEM images of Nb/gradient-Mg:Ta₃N₅/La:Ta₃N₅ film. **b** Cross-sectional STEM image and corresponding EDS elemental mappings for Nb/gradient-Mg:Ta₃N₅/La:Ta₃N₅ film. **c** UPS spectra of the gradient-Mg:Ta₃N₅ and La:Ta₃N₅ films. **d** Band diagrams of gradient-Mg:Ta₃N₅ and La:Ta₃N₅ films determined from UPS and UV-vis absorption measurements. CB conduction band, VB valence band, NHE normal hydrogen electrode. **e** Band bending schematics of the Nb/gradient-Mg:Ta₃N₅/La:Ta₃N₅ photoanode. $E_{F,n}$ and $E_{F,p}$, quasi-Fermi levels; $\Delta E$, built-in electric field.

isostructural replacement of Ta⁵⁺ with La³⁺ in the anosovite structure could negatively affect its crystallinity[34]. Indeed, the diffraction peaks become broader and their intensity gradually decrease with increasing La concentration, suggesting that the incorporation of La into Ta₃N₅ reduced its crystallinity. The grain sizes, obtained by applying the Scherrer equation to the (110) and (203) peaks[35], reduce with increasing La doping concentration (Supplementary Table 2). Furthermore, evident morphology change was observed from the scanning electron microscopy (SEM) images in Fig. 3b–e. With increasing La doping concentration, the porosity of the film increases and the grain size reduces, which agrees with the XRD results.

The grain size of the thin films can potentially impact charge transport properties through trap states at grain boundaries that act as recombination centers[36]. The decrease in average grain sizes leads to the formation of a large number of grain boundaries in the thicker films, resulting in carrier recombination at the grain boundaries. Therefore, the poor PEC performance in the thicker (500 and 700 nm) 3% La-doped Ta₃N₅ photoanodes (Supplementary Fig. 9) may have been caused by the decrease in average grain size with a high La doping level. By reducing the La doping concentration to 2% in the 700 nm film, it is possible to improve the photocurrent density to 9.23 mA cm⁻² at 1.23 V versus RHE (Supplementary Fig. 13). However, the photocurrent is less improved in the low-bias region, indicating poor charge transport in the thick La-doped Ta₃N₅ film without applying a large external bias. It can be seen from Fig. 1b that La doping induces flatter and denser bands near the Fermi level, indicating that the electrons on the band have stronger localization and relatively large effective mass[33]. The effective masses of electrons and holes calculated by fitting the second derivative of the bands are obviously increased after La doping (Supplementary Table 3). This suggests that La-doped Ta₃N₅ may have lower carrier mobilities, resulting in poor charge transport properties in thick films.

The above results indicate that although La doping improves light absorption and reduces Ta³⁺-related trap density in Ta₃N₅, it deteriorates carrier transport due to significantly reduced grain size and increased effective masses of electrons and holes. The positive and negative effects of La doping lead to a performance trade-off in La-doped Ta₃N₅ films prepared with a conventional doping strategy. A better doping strategy is therefore required to decouple the positive and negative effects.

## Combining surface La doping with bulk gradient Mg doping in Ta₃N₅

It has been previously demonstrated that efficient carrier transport in the bulk of Ta₃N₅ film could be achieved by gradient Mg doping[24]. By combining La doping in the surface layer and gradient Mg doping in the bulk, it is expected to decouple the light absorption and carrier transport in Ta₃N₅ thin film. Enhanced light absorption could be achieved with La-doped surface layer, while efficient carrier transport in the bulk is ensured by gradient Mg doping. Hence, a thin La-doped Ta₃N₅ (La:Ta₃N₅) layer was introduced onto gradient Mg-doped Ta₃N₅ (gradient-Mg:Ta₃N₅) film to further improve its photocurrent.

The cross-sectional SEM images in Fig. 4a show the prepared gradient-Mg:Ta₃N₅/La:Ta₃N₅ film on Nb substrate with a thickness of ~650 nm. A compact La:Ta₃N₅ layer with a thickness of ~100 nm is observed on top of the porous gradient-Mg:Ta₃N₅ layer. The high-resolution transmission electron microscopy (HRTEM) images in Supplementary Fig. 14 reveal that the two layers have different crystallinity. The clear lattice fringes of the gradient-Mg:Ta₃N₅ layer without highly distorted grain boundaries indicate high crystallinity inside the film, while the surface La:Ta₃N₅ layer shows lower crystallinity. Figure 4b displays the scanning transmission electron microscopy (STEM) image and corresponding EDS mappings of Nb/gradient-Mg:Ta₃N₅/La:Ta₃N₅ film. The EDS mapping of Mg element reveals a

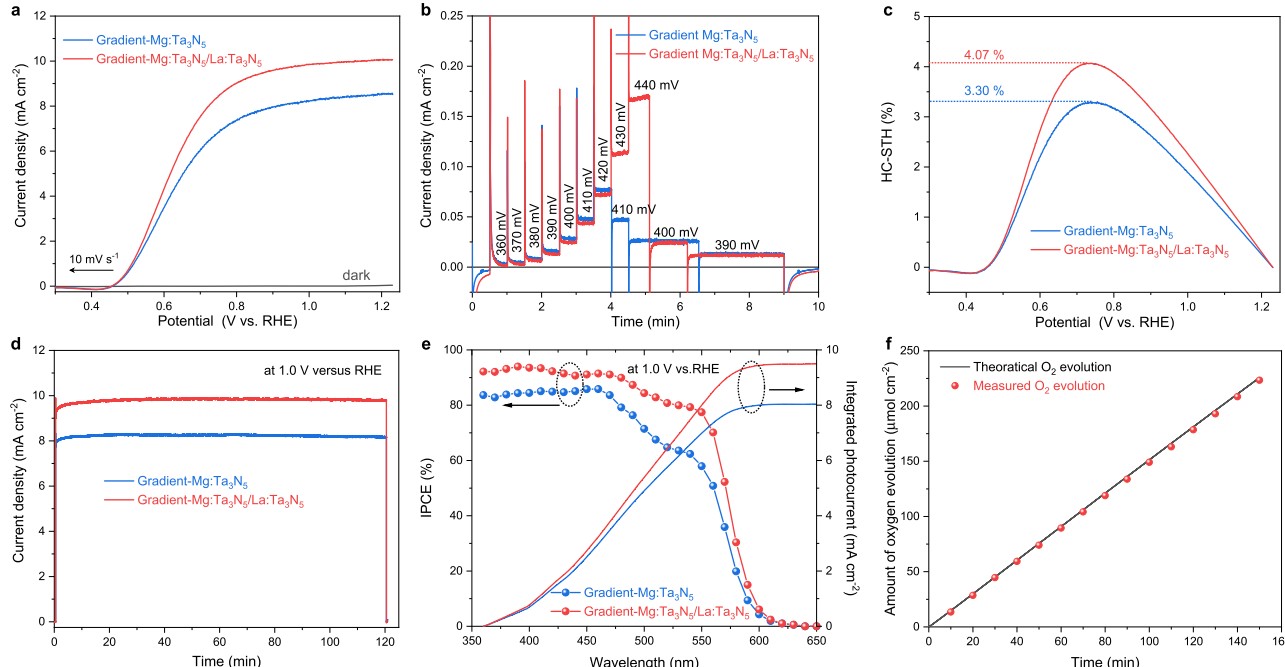

**Fig. 5 | Solar-driven PEC water oxidation properties of gradient-Mg:Ta₃N₅/La:Ta₃N₅ photoanode.** a *J−V* curves for gradient-Mg:Ta₃N₅ and gradient-Mg:Ta₃N₅/La:Ta₃N₅ photoanodes with NiCoFe-Bᵢ co-catalyst in 1 M KOH under AM 1.5G simulated sunlight. **b** Steady-state photocurrent of gradient-Mg:Ta₃N₅/La:Ta₃N₅ photoanode under low applied potentials. **c** HC-STH of the photoanodes calculated from *J−V* curves in (**a**). **d** Steady-state photocurrent for gradient-Mg:Ta₃N₅ and gradient-Mg:Ta₃N₅/La:Ta₃N₅ photoanodes with NiCoFe-Bᵢ co-catalyst at 1.0 V versus

RHE under AM 1.5 G simulated sunlight. **e** IPCE spectrum of gradient-Mg:Ta₃N₅ and gradient-Mg:Ta₃N₅/La:Ta₃N₅ photoanode at 1.0 V versus RHE and the corresponding estimated solar photocurrent over the standard AM 1.5G solar spectrum. **f** Amount of oxygen evolved from the gradient-Mg:Ta₃N₅/La:Ta₃N₅ photoanode under an applied potential of 1.0 V versus RHE. Theoretical O₂ evolution (black line) is calculated from the photocurrent by assuming 100% FE.

gradient distribution of Mg dopants in the Mg:Ta₃N₅ layer. However, the La dopants in the surface layer cannot be directly detected by EDS mapping due to its low concentration. The homogenous mapping result of La is resulted from noise spectra of the EDS instrument. To confirm the existence of La dopants on the surface of the film, an EDS line scan was carried out in the surface region. As shown in Supplementary Fig. 15, the plotted counts of Ta and La elements reveal the presence of low concentration of La dopants in the top 100 nm of the film. The XPS depth profiles further confirm the doping of La in the surface layer (Supplementary Fig. 16). Furthermore, it is revealed that besides the gradient distribution of Mg dopants in the Mg:Ta₃N₅ layer, they may diffuse into the surface layer during the high-temperature nitridation process.

The energy band alignment of the homojunction formed between the La:Ta₃N₅ and gradient-Mg:Ta₃N₅ layers was analyzed using ultraviolet photoelectron spectroscopy (UPS) (Fig. 4c). By subtracting the cut-off energies for the secondary electrons from the He I excitation energy (21.22 eV), the Fermi levels ($E_F$) of La:Ta₃N₅ and gradient-Mg:Ta₃N₅ are calculated to 3.64 and 3.84 eV below the vacuum level, respectively. The UPS spectra in the valence band region reveal that the valence bands of La:Ta₃N₅ and gradient-Mg:Ta₃N₅ are 1.95 and 1.93 eV below their Fermi levels, respectively. Combined with their optical band gaps of ~2.1 eV, the energy band positions of the La:Ta₃N₅ and gradient-Mg:Ta₃N₅ layers were plotted in Fig. 4d. When the gradient-Mg:Ta₃N₅/La:Ta₃N₅ homojunction is formed and in contact with the electrolyte, the Fermi levels equilibrate through the redistribution of free carriers resulting in band bending, as shown in Fig. 4e. A built-in electric field of ~0.2 eV at the interface between La:Ta₃N₅ and gradient-Mg:Ta₃N₅ is created, which enhances charge separation in the near-surface region. Meanwhile, the gradient Mg doping in the inner layer induces a gradient band structure, which enhances the bulk charge separation efficiency[24]. The band structure of the gradient-Mg:Ta₃N₅/La:Ta₃N₅ homojunction could allow for efficient hole transfer to the

electrolyte and electron transfer to the substrate. Combining with the enhanced light absorption, it is expected this homojunction photoanode could achieve improved PEC performance.

The charge separation by the homojunction was probed by measuring the steady-state PL and TRPL spectra of gradient-Mg:Ta₃N₅ and gradient-Mg:Ta₃N₅/La:Ta₃N₅ films, as shown in Supplementary Fig. 17. Both samples display a near-band-edge emission peak centered at ~590 nm. It is noted that the PL intensity of the gradient-Mg:Ta₃N₅/La:Ta₃N₅ film was significantly quenched with respect to that of the gradient-Mg:Ta₃N₅ film. The PL quenching was ascribed to the efficient charge transfer from the gradient-Mg:Ta₃N₅ to the La:Ta₃N₅ layer, in addition to the effective exciton dissociation which suppresses the photogenerated carriers recombination[37,38]. The TRPL measurements showed corroborating results. The average lifetimes of the biexponential decay were fitted to be 1.00 and 0.61 ns for gradient-Mg:Ta₃N₅ and gradient-Mg:Ta₃N₅/La:Ta₃N₅ films, respectively (Supplementary Fig. 17b). The decreased lifetime indicated the efficient extraction and transport of the photogenerated charge carriers.

The gradient-Mg:Ta₃N₅/La:Ta₃N₅ films deposited on Nb substrates were used for PEC measurements. To enhance oxygen evolution kinetics, a NiCoFe-Bᵢ oxygen evolution co-catalyst layer was deposited on the surface of the photoelectrodes[39]. SEM images (Supplementary Fig. 18) revealed that the gradient-Mg:Ta₃N₅/La:Ta₃N₅ films were conformally covered with a NiCoFe-Bᵢ layer. Moreover, photoelectrochemical impedance spectroscopy analysis reveals that the charge transfer resistance across the photoanode/electrolyte interface is significantly reduced after NiCoFe-Bᵢ co-catalyst modification (Supplementary Fig. 19 and Supplementary Table 4). This indicates that the co-catalyst modification is necessary to improve the charge transfer kinetics across the photoanode/electrolyte interface[40,41]. Figure 5a shows the *J−V* curves of gradient-Mg:Ta₃N₅ and gradient-Mg:Ta₃N₅/La:Ta₃N₅ photoanodes measured in the dark and under air mass (AM)

1.5G-simulated sunlight. To exclude potential influence of the film thickness, the PEC properties of the gradient-Mg:Ta₃N₅/Ta₃N₅ photoanode are also measured for comparison, as shown in Supplementary Fig. 20. The gradient-Mg:Ta₃N₅ photoanode achieves a photocurrent density of 8.51 mA cm⁻² at 1.23 V versus RHE, which is consistent with our previous report[24]. The gradient-Mg:Ta₃N₅/Ta₃N₅ photoanode showed a decreased photocurrent density of 8.2 mA cm⁻² at 1.23 V versus RHE. An apparent increase of the saturation photocurrent was observed for the gradient-Mg:Ta₃N₅/La:Ta₃N₅ photoanode, yielding a remarkable photocurrent density of 10.06 mA cm⁻² at 1.23 V versus RHE. The statistics of the J−V curves for 25 gradient-Mg:Ta₃N₅/La:Ta₃N₅ photoanodes prepared under the same conditions shows that photocurrent density is 9.92 ± 0.24 mA cm⁻² at 1.23 V versus RHE, corresponding to an enhancement of ~18% compared to 8.43 ± 0.22 mA cm⁻² for the gradient-Mg:Ta₃N₅ photoanode (Supplementary Fig. 21).

The steady-state photocurrent at low bias conditions was measured in Fig. 5b to determine the onset potential of the gradient-Mg:Ta₃N₅/La:Ta₃N₅ photoanode. A steady photocurrent density of ~20 μA cm⁻² was generated at 0.39 V versus RHE, revealing a low onset potential similar to that of the gradient-Mg:Ta₃N₅ photoanode. However, due to the significantly increased saturation photocurrent, the gradient-Mg:Ta₃N₅/La:Ta₃N₅ photoanode yielded a maximum HC-STH of 4.07% (Fig. 5c), as compared with the 3.30% for the gradient-Mg:Ta₃N₅ photoanode. It is noted that this is the highest HC-STH ever reported for Ta₃N₅-based photoanode[42], to the best of our knowledge (Supplementary Fig. 22). It also exceeds the highest HC-STH reported for α-Fe₂O₃ (ref. [43]) and BiVO₄ (ref. [44]) based single-photon photoanodes by a considerable margin. Statistically, the gradient-Mg:Ta₃N₅/La:Ta₃N₅ photoanodes achieved an average HC-STH of 3.92% with a small standard deviation of 0.08% (Supplementary Fig. 21c). The stability of the gradient-Mg:Ta₃N₅ and gradient-Mg:Ta₃N₅/La:Ta₃N₅ photoanodes in 1 M KOH was tested under continuous simulated sunlight at 1.0 V versus RHE. The steady-state photocurrent densities in Fig. 5d match those of the J−V curves at the same potential in Fig. 5a and both samples showed a stable photocurrent for 120 min.

Figure 5e plots the wavelength dependence of the incident photon-to-current conversion efficiency (IPCE) for the gradient-Mg:Ta₃N₅ and gradient-Mg:Ta₃N₅/La:Ta₃N₅ photoanodes measured at 1.0 V versus RHE. Both samples show photocurrent response to incident light in the wavelength region below 600 nm, which matches the bandgap of Ta₃N₅ (~2.1 eV). The IPCE of the gradient-Mg:Ta₃N₅ photoanode exceeds 80% in the wavelength range from 360 to 470 nm, but it gradually decreases from 79.2% at 480 nm to 57.9% at 550 nm. The obvious decrease of IPCE in this spectral range reveals the negative effect of optical anisotropy in Ta₃N₅ which lowers the light absorption efficiency. In contrast, the IPCE of the gradient-Mg:Ta₃N₅/La:Ta₃N₅ photoanode shows a decrease to a lesser extent from 89.7% at 480 nm to 77.4% at 550 nm. The improved IPCE in this spectral range is mainly ascribed to the enhanced light absorption by La doping that alleviates the effect of optical anisotropy. By multiplying the IPCE spectra with the standard AM 1.5G spectrum (ASTM G173-03), the solar photocurrent spectra were calculated (Supplementary Fig. 23). Comparing the solar photocurrent spectra of the two samples, it is obvious that the most significant enhancement of photocurrent comes from the spectral range of 480–550 nm. The integrated solar photocurrent densities are 8.05 and 9.50 mA cm⁻² for the gradient-Mg:Ta₃N₅ and gradient-Mg:Ta₃N₅/La:Ta₃N₅ photoanodes, respectively. These values match well with the photocurrent densities measured from the J−V curve at 1.0 V versus RHE in Fig. 5a. The oxygen evolved from the photoanode at 1.0 V versus RHE under AM 1.5 G was quantified by gas chromatography (Fig. 5f). The continuous production of oxygen with Faradaic efficiency (FE) close to 100% demonstrates that the high and stable photocurrent (Supplementary Fig. 24) is mainly utilized for oxygen evolution rather than for other side reactions.

## Discussion

In summary, we demonstrated a heterogeneous doping strategy to decouple light absorption and carrier transport in Ta₃N₅ thin film photoanode. We found that La doping could effectively increase light absorption in the wavelength range of 480–590 nm by tuning the optical anisotropy of Ta₃N₅ thin film. However, the carrier transport in the bulk of the film is hindered by La doping due to significantly reduced grain size and increased effective masses of electrons and holes. To break performance trade-offs caused by these factors, La doping in the surface layer was combined with gradient Mg doping in the bulk of the Ta₃N₅ thin film. This heterogeneous doping strategy led to enhanced light absorption and carrier transport. In addition, studies of band edge energetics revealed the formation of gradient-Mg:Ta₃N₅/La:Ta₃N₅ homojunction with build-in electric field, which further improved the charge separation efficiency. As a result, the heterogeneously doped Ta₃N₅ thin film photoanode achieved a remarkable photocurrent density of 10.06 mA cm⁻² at 1.23 V versus RHE, a low onset potential of 0.39 V versus RHE, and a maximum HC-STH of 4.07%. This work demonstrates heterogeneous doping is an effective strategy to break performance trade-offs by decoupling light absorption and carrier transport in semiconductor thin film light absorbers.

## Methods

### Materials preparation

The oxide precursor films were deposited on niobium (Nb) foils (99.99% in purity, 10 × 10 × 0.1 mm³ in size) and quartz glass substrates by dual-source electron-beam evaporation. Ta₂O₅ (99.99% in purity), La₂O₃ (99.99% in purity), and MgO (99.99% in purity) were used as the Ta, La and Mg sources, respectively. The deposition rate was controlled by adjusting the source power and monitored by quartz crystal microbalance (QCM). La-doped Ta₂O₅ films with different doping contents were prepared by fixing the deposition rate of Ta₂O₅ at 5 Å s⁻¹, while changing the deposition rate of La₂O₃ from 0.15 to 0.3 Å s⁻¹. The preparation of films with gradient Mg doping concentration was based on our previously reported procedure. The deposition rate of Ta₂O₅ was fixed at 5 Å s⁻¹, while that of MgO was varied linearly from 0.9 to 0.4 Å s⁻¹. After the deposition, the as-prepared oxide precursor films were then converted to cation doped Ta₃N₅ films through a one-step nitridation process under an NH₃ flow of 150 sccm at 1273 K for 6 h at a ramp rate of 10 K min⁻¹. For comparison, undoped Ta₃N₅ films were prepared by directly nitriding Ta₂O₅ films in the same manner as described above.

### Materials characterization

The crystal structures of the films were characterized using XRD (Thermo Scientific ARL EQUINOX 1000) with Cu Kα operated at 40 kV and 30 mA. The film morphology and composition were characterized using a Zeiss NVision 40 field-emission SEM associated with an EDS (Oxford Ultim Max 40). The absorbance was measured using an UV-vis spectrophotometer (Ocean Optics QE Pro). The low-temperature PL spectra were measured using a PL spectrometer (Picoquant FluoTime 300) by cooling the sample to -10 K in a closed-cycle He cryostat (ARS DE-202). TRPL spectra were acquired using the same PL spectrometer with a 420 or 510 nm picosecond laser pulsed at a repetition rate of 40 MHz as the excitation sources. XPS (Thermo Scientific ESCALAB 250 Xi) was carried out using Al Kα source and the binding energy was calibrated by setting the binding energy of the hydrocarbon C 1 s peak at 284.8 eV. XPS depth profiles were conducted by etching the samples with 4 keV argon ion beam and a raster size of 2 × 2 mm². Quantitative XPS analyses were performed using Thermo Scientific Avantage Software. The UPS measurements were performed using the same XPS system with a He I (21.22 eV) excitation line. For cross-sectional STEM measurements, the samples were prepared by focused ion beam (JEOL JIB-4600F) etching or by Ar ion milling using an Ion Slicer (JEOL EM-09100IS) and a Precision Ion Polishing System (Gatan Model 691) for

finishing. The cross-sectional STEM and HRTEM images and EDS mapping were taken with a JEOL JEM-2800.

## Theoretical calculations

The DFT calculations were performed using the Vienna Ab initio Simulation Package (VASP) code with the projector augmented-wave (PAW) method[45]. To optimize the crystal structure, the plane-wave cutoff energy for basic functions was set to 550 eV. The exchange-correlation functional of the generalized gradient approximation (GGA) of the Perdue-Burke-Ernzerh (PBE)[46] was used. The Hellmann-Feynman forces on ions were below 0.01 eV/Å and accuracy for electronic minimization was $1 \times 10^{-5}$ eV. The Γ-centered $17 \times 7 \times 7$ and $7 \times 5 \times 5$ K-mesh in the Brillouin zone were used on $2 \times 1 \times 1$ supercell models for structural optimization and electronic calculations, respectively. To obtain the accurate electronic properties, the modified Becke-Johnson (MBJ) exchange potential was utilized for the exchange and correlation effect, which can yield band gaps with an accuracy close to hybrid functional[47,48].

## Photoanode fabrication

The samples deposited on Nb substrates were used to prepare the photoanodes. The back side of the substrate was connected to a copper wire by soldering with indium, then encapsulated with epoxy (Araldite). The exposed area of the electrode after packaging was measured with calibrated digital images and ImageJ[1.53k] software (Supplementary Fig. 25). For co-catalyst modification, NiCoFe-B$_i$ co-catalyst was deposited on the photoanodes using a photo-assisted electrodeposition method. Before co-catalyst deposition, the samples were dipped in a mixed etchant of HF:HNO$_3$:H$_2$O (1:2:7 in v/v) for 20 s to prepare a fresh surface, and then rinsed with deionized water. Subsequently, the electrodes were transferred to a stirred and Ar-purged solution containing 2 mM NiSO$_4$·6H$_2$O (99.99% metals basis, Aladdin), 0.5 mM Co(NO$_3$)$_2$·6H$_2$O (99.99% metals basis, Aladdin) and 0.8 mM FeSO$_4$·7H$_2$O (99.95% metals basis, Aladdin) in 0.25 M potassium borate (K$_2$B$_4$O$_7$·4H$_2$O) buffer at pH 10. The photo-assisted electrodeposition was performed in a three-electrode configuration with Ag/AgCl as the reference electrode and a Pt wire as the counter electrode. The NiCoFe-B$_i$ co-catalyst was electrodeposited onto the electrodes at a constant current density of 30 μA cm$^{-2}$ for 10 min under AM 1.5G simulated sunlight. After the deposition, the electrodes were rinsed with deionized water.

## PEC measurements

All PEC measurements were conducted on a potentiostat (BioLogic SP-200) in three-electrode configuration using a Pt cathode and a Hg/HgO reference electrode. To prevent the back reaction and the light scattering effect of the generated H$_2$ bubbles, the Pt cathode chamber and the photoanode chamber were separated using a Nafion 117 membrane. A class AAA solar simulator (SAN-EI ELECTRIC, XES-40S3-TT) was used as the light source, and the irradiance was adjusted to 100 mW cm$^{-2}$ using a certified reference cell (Konica-Minolta AK-200). The temperature of the electrolyte (1 M KOH, pH 13.6) was maintained at 283 K using a constant temperature water bath during the PEC test. Linear sweep voltammogram ($J$–$V$ curves) were recorded under a cathodic scan at a rate of 10 mV s$^{-1}$. All the measured potentials versus Hg/HgO reference electrode were converted to the potentials versus RHE according to the Nernst equation. The HC-STH was calculated from the $J$–$V$ curves under AM1.5 G illumination using the equation: HC-STH = $[J \times (1.23 - V_{app})/P_{light}] \times 100$%, where $V_{app}$ is the applied potential versus RHE, $J$ is the photocurrent density under AM 1.5G light and $P_{light}$ is the irradiance of the simulated sunlight (100 mW cm$^{-2}$). Chronoamperometry measurements (Steady-state photocurrent curves) were carried out at a potential of 1.0 V versus RHE under AM 1.5G simulated sunlight (100 mW cm$^{-2}$). Mott-Schottky measurements were conducted by performing an anodic potential scan under frequencies of 0.5, 1.0, and 1.5 kHz with an AC

amplitude of 15 mV under the dark conditions. The IPCE spectra were measured using a monochromatic light source (Zolix Sirius 300P) in the wavelength range from 360 to 650 nm with a 10 nm interval at 1.0 V versus RHE in 1 M KOH. The intensity of the monochromatic light was measured using a calibrated reference cell (Thorlabs PDS1010-CAL). The IPCE at each wavelength ($\lambda$) was calculated by: IPCE = $[(1240/\lambda) \times (J_{light} - J_{dark})/P_{light}] \times 100$%, where $P_{light}$ is the intensity of the monochromatic light, $J_{light}$ and $J_{dark}$ are the photocurrent density under illumination and in the dark, respectively. The amounts of O$_2$ gases evolved from the photoanode in a closed circulation PEC system were analyzed by a gas chromatograph (Shimadzu GC-2014) equipped with a thermal conductivity detector (TCD) and a packed column using high purity Ar as a carrier gas.

## Data availability

Source data are provided with this paper.

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

## Acknowledgements

This work was supported by the National Natural Science Foundation of China (No. 21872019). M.N., N.S., and K.D. acknowledge the Artificial Photosynthesis Project (ARPChem) of the New Energy and Industrial Technology Development Organization (NEDO). A Part of this study was supported by the University of Tokyo Advanced Characterization Nanotechnology Platform in the Nanotechnology Platform Project sponsored By the Ministry of Education, Culture, Sports, Science and Technology (MEXT), Japan (JPMXP09-A-20-UT-0004).

## Author contributions

Y.L. and Y.X. conceived the idea; Y.L., N.S., and K.D. supervised the project. Y.X. carried out the device fabrication and PEC tests with the assistance from Z.F. and C.L.; M.N. performed the STEM and HRTEM experiments; Q.L. and L.Z. performed the DTF calculations. Y.L., Y.X., Q.W., and K.D. analyzed the results and wrote the manuscript. All authors commented on and revised the manuscript.

## Competing interests

The authors declare no competing interests.
