## [Peer Review File · Nature Communications]

Decoupling light absorption and carrier transport via heterogeneous doping in Ta₃N₅ thin film photoanodeREVIEWER COMMENTS

Reviewer #1 (Remarks to the Author):

See attachment

In this article, titled, “Decoupling light absorption and carrier transport via heterogeneous doping in Ta₃N₅ thin film photoanode” the authors synthesis Ta₃N₄ photoanodes that contain a gradient of co-doping, with higher levels of La doping at the surface and higher levels of Mg in the bulk. This creates a homojunction that the authors claim decouples light absorption and charge transport in the photoanode, enabling some of the highest solar to hydrogen (STH) efficiencies reported in this material and related photoanodes to date. The materials are well characterised by a range of physical and photoelectrochemical methods. Overall, I find the manuscript to be of a high standard and impact, and I recommend its publication in your journal after the minor concerns I have presented have been addressed.

Typos/ grammatical errors

- 1) Line 42: “is therefore demanded” should be “is therefore required”
- 2) Line 59: “in Ta₃N₅ thin film” should be “in a Ta₃N₅ thin film”
- 3) Line 159: “indicating the lattice expansion” should be “indicating a lattice expansion”

Queries

- 1) **Abstract:** “a record-high applied bias photon-to-current efficiency of 4.07%” should be “a record-high solar-to-hydrogen efficiency of 4.07%”

The name, applied bias photon-to-current efficiency is a misnomer. You are calculating the STH at various applied potentials. See Equation 2 in Chem. Rev. 2010, 110, 11, 6446–6473

2) In the introduction section, you discuss the loss of photovoltage that may occur in nanostructured devices from the decrease in light absorption per unit area of material. You state on line 45 that “this is not achieved without sacrificing the photovoltage of the semiconductor liquid junction”. However, it should be noted that there are numerous examples in the literature where the photocatalytic onset potential of the photoelectrode does not change when going from a thin film to a nanostructured material (e.g. Cu₂O, Nano Lett. 2016, 16, 3, 1848–1857; WO₃, J. Phys. Chem. C 2017, 121, 11, 5983–5993; etc). Therefore, your statement should be toned down (as what should theoretically happen doesn’t always happen in practice). Nevertheless, I suggest that another potential benefit of using a thin film, in that it will likely be more durable than a nanostructured material, and also less susceptible to corrosion, can be discussed.

3) Could you please show the original absorption spectra of both the undoped and La doped material in Figure 1 in the SI (plot side by side). Figure S1b is a tad confusing, but can be kept so long as what I ask for is also provided.

4) On line 105, you write “As shown in Fig. 1e, the undoped Ta₃N₅ with intrinsic ON, VN and Ta³⁺ defects shows high density of intermediate states, which is consistent with our previous results.” I understand that this is indeed previously published work, but can you provide some brief details to the reader on what orbitals (and their associated ions) contribute to the intermediate states seen in the doped and undoped systems.

5) On line 117 you write, “Quantitative XPS analyses show that the La/Ta ratio in the La-doped Ta₃N₅ thin film is 2.97%.” Can the authors comment on the synthesis and if this was the expected surface doping level.

6) In Figure S7, you write in the caption for c and d that the applied potential was 1.0 V vs RHE; however, in the figures themselves you have written 1.23 V vs RHE. Please clarify. Also, you state that samples were measured under AM 1.5G radiation. Although it is obvious, please make it clear to the reader that this was using a solar simulated light source.

7) On line 171 you write, "The significantly reduced grain size leads to severe carrier recombination at the grain boundaries". This statement is too conclusive. Instead you should infer that this may have been caused by the decrease in average crystal size displayed in samples with higher levels of La doping.

8) The data for undoped Ta₃N₅ at various thicknesses shown in Figure S7 is not discussed in the main text. Please provide reasoning to the reader for why the 700 nm thick Ta₃N₅ shows a higher photocurrent density than that of any La-doped Ta₃N₅ photoanode produced herein.

9) On line 184, you write, "The above results show that although La doping improves light absorption..." Again, the statement "show" is too strong. Rather, your data "indicates" this.

10) I recommend that authors introduce a table that summarises the PEC data of all samples produced herein. This can go in the SI. One way of presenting this data could be the photocurrent density at a specific RHE. The doping level can be on the top row and the columns on the left can be the film thickness. You can also do this for maximum "ABPE" (or preferably STH, as I state in point 1) and the photocatalytic onset potential if you wish also.

11) On line 209 you write, "However, the La dopants in the surface layer cannot be directly detected by EDS mapping due to its low concentration." Also, Figure 4a implies that there is no Mg doping at the surface of the material, and that is solely lies in the material bulk.

To clarify what the surface and bulk constituents are in your material I strongly recommend XPS analysis of the surface and bulk (through sputtering) of your La and Mg co-doped Ta₃N₅ material. I recommend you investigate La, Mg, Ta, N and O content.

12) Figure 16d should also be produced for the Mg-gradient doped Ta₃N₅ samples shown in Figure 16c, for comparison. Personally, it would be better if Figure 16b and c were merged. And similarly, when constructing this new figure I have now requested, it be merged with Figure 16d also.

13) On line 262 you write "1.0 V versus RHE" but again the figure itself writes 1.23 V vs RHE. Please clarify.

Reviewer #2 (Remarks to the Author):

This paper uses a heterogeneous doping strategy to improve the photoabsorption properties of Ta₃N₅ at photoanode surface (La:Ta₃N₅) while preserving the efficient carrier transport properties of the Mg:Ta₃N₅ thin film, in doing so achieving a very high ABPE efficiency of 4.07%, a record for Ta₃N₅ based photoanodes. The mechanism of increased photoabsorption through La doping is attributed to reduced optical anisotropy, or in other words, slightly disrupting the Ta₃N₅ lattice. The La³⁺ doping is also claimed to improve the charge transfer properties of Ta₃N₅ films, notably only compared to bare Ta₃N₅ film rather than other doped Ta₃N₅ films like Mg:Ta₃N₅, by the introduction of ON donors and the suppression of VN and Ta³⁺ defects when the concentration of La³⁺ is not too high (~3%), otherwise the charge transfer properties suffer when the lattice structure of Ta₃N₅ is greatly distorted. The charge transfer performance is further aided by using Mg:Ta₃N₅ on the back of La:Ta₃N₅.

This paper introduces a novel way of reducing optical anisotropy by "lattice engineering" via doping, which is industrially relatively easy to implement, and shows up to 100% more photoabsorption at wavelengths close to bandgap, and consistently improved photoabsorption at photoenergies greater than bandgap. The performance of the La:Ta₃N₅ film developed to fulfill such goal was thoroughly measured, including honestly admitting that the charge transfer properties of such film suffers from disturbed lattice structure, and the charge transfer properties of La:Ta₃N₅ is considerably worse than that of Mg:Ta₃N₅, or even Ta₃N₅ in some cases. This paradox of light absorption and charge transfer calls for optimization of La dopant concentration and La:Ta₃N₅ film thickness, which the authors carried out systematically. Thus, engineering wise, the authors explored a lot of parameter space and provided enough information to validate the effectiveness of their photoanode.

However, despite showing the XPS survey spectra of La:Ta₃N₅ and Ta₃N₅, the authors did not explicitly specify why La is the go to dopant for tuning optical anisotropy of Ta₃N₅, and what are the shortcomings of other dopant candidates. From our understanding, La doping have been used in La:STO systems for decades for its electrochemical properties, i.e. reduce electron concentration, and La:STO coating have already been introduced to the photocatalysis community (<https://doi.org/10.1016/j.solmat.2020.110428>). This begs the question, what are the significant differences, electrochemical mechanism wise, between La doping in La:STO and La:Ta₃N₅? Is La doping a universal solution to increase photoabsorption performance across all common photoanode materials? Additionally, the electrochemical performance benchmark for La:Ta₃N₅ to compare to should not only be Ta₃N₅, but also Mg:Ta₃N₅, so the XPS data of Mg:Ta₃N₅ should be added as well, along with a small discussion of light absorbing properties of Mg:Ta₃N₅.

In addition to the properties of the La:Ta₃N₅ coating itself, when Ni based Oxygen evolving catalyst is conformally introduced to the coating, the authors should give a brief discussion about the interface between catalyst and the coating, i.e. the transportation properties of charge carriers.

Reviewer #3 (Remarks to the Author):

As one of the most promising photoanode materials for solar-driven photoelectrochemical (PEC) water splitting, tantalum nitride (Ta₃N₅) has received tremendous research attention in recent years. A steady improvement in the applied bias photon to current efficiency (ABPE) has been achieved through different strategies such as nanostructure engineering, defects engineering, and interface engineering. Here, Li et al. report a heterogeneous doping strategy to decouple the light absorption and carrier transport in Ta₃N₅ thin-film photoanode. A maximum ABPE of 4.07% is obtained for PEC water splitting, which to the best of my knowledge, is the highest among reported values for Ta₃N₅ photoanodes. The result is significant and very exciting, especially the effect of La doping on alleviating anisotropic optical absorption in Ta₃N₅, which was neglected in previous studies of Ta₃N₅ photoanodes. The proposed heterogeneous doping strategy could also serve as a guideline for improving the light conversion efficiency of other semiconductor thin films by breaking performance

trade-offs between light absorption and carrier transport. Overall, this study is very interesting and the paper is well-articulated. The paper can be published after addressing the following minor remarks:

1. If I understood correctly, the term "pristine Ta₃N₅" refers to an ideal Ta₃N₅ crystal without any defects while "undoped Ta₃N₅" refers to Ta₃N₅ with intrinsic defects such as nitrogen vacancies and reduced Ta³⁺ species. The correct use of the terms should be checked throughout the manuscript to avoid confusion.
2. For the valence band positions of La-doped Ta₃N₅ film, why do the values obtained by XPS valence band spectra (Fig. S6) and USP spectra (Fig. 4c) differ significantly?
3. The pH condition for the potential scale of the normal hydrogen electrode in Figure 4d should be given.
4. Fig. 5a shows that the gradient-Mg:Ta₃N₅/La:Ta₃N₅ photoanode has a higher photocurrent than that of the gradient-Mg:Ta₃N₅ photoanode. Since the two samples have different thicknesses, it is better to conduct a control experiment to compare samples with the same Ta₃N₅ film thickness. For instance, the surface La-doped Ta₃N₅ layer can be replaced with an undoped Ta₃N₅ layer having the same thickness on the gradient Mg-doped samples for comparison.
5. The abbreviation "FE" in line 592 was not defined before use.

Point-by-Point Responses to Reviewers' Comments

Manuscript number: NCOMMS-22-37115-T

Manuscript type: Research Article

Title: "Decoupling light absorption and carrier transport via heterogeneous doping in Ta₃N₅ thin film photoanode"

Correspondence Authors: Prof. Yanbo Li

Authors: Yequan Xiao, Mr Zeyu Fan, Mamiko Nakabayashi, Qiaoqiao Li, Prof. Liujiang Zhou, Prof. Qian Wang, Prof. Changli Li, Prof. Naoya Shibata, Prof. Kazunari Domen

Dear Reviewers:

We appreciate the referee's careful review and valuable comments and suggestions. We have revised the manuscript thoroughly according to those comments. The reviewer comments are shown in *italic*; our responses are in **blue**; the added items and revisions are highlighted in **red** in the main text and the supplementary information. Our point-by-point responses to the reviewers' comments are provided below:

Reviewer #1

General comment: In this article, titled, "Decoupling light absorption and carrier transport via heterogeneous doping in Ta₃N₅ thin film photoanode" the authors synthesis Ta₃N₅ photoanodes that contain a gradient of codoping, with higher levels of La doping at the surface and higher levels of Mg in the bulk. This creates a homojunction that the authors claim decouples light absorption and charge transport in the photoanode, enabling some of the highest solar to hydrogen (STH) efficiencies reported in this material and related photoanodes to date. The materials are well characterised by a range of physical and photoelectrochemical methods. Overall, I find the manuscript to be of a high standard and impact, and I recommend its publication in your journal after the minor concerns I have presented have been addressed.

General Response: We greatly appreciate that the reviewer finds our manuscript to be of a high standard and impact, and thank the reviewer for recommending our work for publication. We further would like to thank the reviewer for the constructive comments which helped to further improve the

quality of the paper. In the revised manuscript, we have considered all points raised and provided point-by-point responses to the reviewer's comments. We hope that the details provided below will address the reviewer's concerns.

Typos/ grammatical errors

- 1) Line 42: "is therefore demanded" should be "is therefore required"
- 2) Line 59: "in Ta₃N₅ thin film" should be "in a Ta₃N₅ thin film"
- 3) Line 159: "indicating the lattice expansion" should be "indicating a lattice expansion"

Response: Thank the reviewer very much for pointing out these grammatical errors. We are sorry for these mistakes and all the format and grammatical errors have been corrected. The corresponding corrections are on **lines 42, 60, and 174** of the revised manuscript.

Queries

Comment 1: Abstract: "a record-high applied bias photon-to-current efficiency of 4.07%" should be "a record-high solar-to-hydrogen efficiency of 4.07%" The name, applied bias photon-to-current efficiency is a misnomer. You are calculating the STH at various applied potentials. See Equation 2 in Chem. Rev. 2010, 110, 11, 6446-6473.

Response 1: We thank the reviewer for this valuable suggestion and have replaced "applied bias photon-to-current efficiency (ABPE)" with "half-cell solar-to-hydrogen conversion efficiency (HC-STH)" in the revised manuscript.

Comment 2: In the introduction section, you discuss the loss of photovoltage that may occur in nanostructured devices from the decrease in light absorption per unit area of material. You state on line 45 that "this is not achieved without sacrificing the photovoltage of the semiconductor liquid junction". However, it should be noted that there are numerous examples in the literature where the photocatalytic onset potential of the photoelectrode does not change when going from a thin film to a nanostructured material (e.g. Cu₂O, Nano Lett. 2016, 16, 3, 1848-1857; WO₃, J. Phys. Chem. C 2017, 121, 11, 5983-5993; etc). Therefore, your statement should be toned down (as what should theoretically happen doesn't always happen in practice). Nevertheless, I suggest that another potential benefit of using a thin film, in that it will likely be more durable than a nanostructured material, and also less susceptible to corrosion, can be discussed.

Response 2: We thank the reviewer for this valuable suggestion. The statement of possible photovoltage losses in nanostructured devices is mainly based on the Shockley diode equation

(Osterloh, *Chem. Soc. Rev.*, **2013**, *42*, 2294-2320). We agree with the reviewer that such a statement is too strong. We have revised this statement in the introduction according to the reviewer's suggestion. Some description of the potential advantages of thin-film photoelectrodes has also been briefly added to the introduction, as follows:

On lines 44-49: "However, the increased junction area in nanostructured photoelectrodes decreases the photon flux received by the semiconductor per unit area, which in theory may reduce the attainable photovoltage of the semiconductor liquid junction¹¹. Consequently, the thermodynamic driving force for water splitting is lowered and the photocurrent onset potential of the nanostructured photoelectrode is negatively affected."

On lines 55-57: "Moreover, the thin-film photoelectrodes are likely more durable and less susceptible to corrosion than nanostructured photoelectrodes, and more suitable for scaling up."

Reference

11. Osterloh, F. E. Inorganic nanostructures for photoelectrochemical and photocatalytic water splitting. *Chem. Soc. Rev.* **42**, 2294-2320 (2013).

Comment 3: Could you please show the original absorption spectra of both the undoped and La doped material in Figure 1 in the SI (plot side by side). Figure S1b is a tad confusing, but can be kept so long as what I ask for is also provided.

Response 3: Thank you for the valuable suggestion. We have provided the original absorption spectra in **Supplementary Fig. 1a** as you suggested. In addition, we added the UV-vis absorption spectra data for Mg-doped Ta₃N₅ film with the same thickness according to Reviewer #2's comment #4. To avoid confusion, we have removed Supplementary Fig. 1b of the original SI.

The related information has now been included/updated in Supplementary Figs. 1 and 6 of the Supplementary Information.

Supplementary Fig. 1 | Optical absorption properties of undoped, La-doped and Mg-doped Ta₃N₅ films on quartz substrates. a, UV-vis absorption spectra. The thickness of all films is about 100 nm. The Mg/Ta concentration in the Mg-doped Ta₃N₅ thin film was 13.5% estimated from XPS results (Supplementary Fig. 6). Compared with the undoped Ta₃N₅ film, the above-bandgap light absorption is weakened in the Mg-doped sample, while that of the La-doped sample is notably enhanced in the range of 480-590 nm. **b**, Tauc plots of UV-vis absorption spectra. α , absorption coefficient; h , Planck's constant; ν , photon's frequency. The dotted lines show the extrapolation of the linear portion of the absorption edges. Mg doping causes an increase in the optical bandgap, while La doping narrows the bandgap of Ta₃N₅ films.

Supplementary Fig. 6 | XPS core-level spectra of Mg-doped Ta₃N₅. a, Mg 1s peak. b, Ta 4f peak. c, O 1s peak. d, N 1s peak. Quantitative XPS analysis showed that the Mg/Ta concentration was approximately 13.5%, and the N/Ta and O/Ta atomic ratios were 1.62 and 0.30, respectively. These results are consistent with our previous studies, which suggested that Mg doping can effectively reduce deep traps created by nitrogen vacancies and increase shallow donors generated by oxygen impurities in Ta₃N₅ (Ref. 1).

Supplementary References

1. Xiao, Y. *et al.* Band structure engineering and defect control of Ta₃N₅ for efficient photoelectrochemical water oxidation. *Nat. Catal.* **3**, 932-940 (2020).

Comment 4: On line 105, you write “As shown in Fig. 1e, the undoped Ta₃N₅ with intrinsic O_N, V_N and Ta³⁺ defects shows high density of intermediate states, which is consistent with our previous results.” I understand that this is indeed previously published work, but can you provide some brief details to the reader on what orbitals (and their associated ions) contribute to the intermediate states seen in the doped and undoped systems.

Response 4: We thank the reviewer for this valuable comment. According to the reviewer's suggestion, we have provided more discussion about the DFT calculations and have added the contribution of different orbitals to the total densities of states (DOS) for undoped Ta₃N₅ and La-doped Ta₃N₅ with intrinsic O_N, V_N, and Ta³⁺ defects in **Supplementary Fig. 4**. The intermediate states in undoped Ta₃N₅ mainly consist of Ta 5*d*, N 2*p* orbitals (**Supplementary Fig. 4a**). Upon doping of La into Ta₃N₅ lattice, the intermediate states mostly originate from Ta 5*d*, N 2*p* orbitals, and a small amount of La 5*d* orbitals (**Supplementary Fig. 4b**). Moreover, the undoped Ta₃N₅ shows a high density of intermediate states, while the density of intermediate states is significantly reduced in La-doped Ta₃N₅.

The related information has now been provided in **Supplementary Fig. 4** and on page 6 of the main text.

Supplementary Fig. 4 | Calculated densities of states (DOS) of undoped Ta₃N₅ (a) and La-doped Ta₃N₅ (b) with intrinsic O_N, V_N, and Ta³⁺ defects. The intermediate states mainly consist of Ta 5*d*, N 2*p* orbitals in undoped Ta₃N₅. Upon doping La into Ta₃N₅ lattice, the intermediate states mostly originate from Ta 5*d*, N 2*p* orbitals, and a small amount of La 5*d* orbital. Also, the density of intermediate states is significantly reduced in La-doped Ta₃N₅.

On lines 111-118: "For the undoped Ta₃N₅ with intrinsic O_N, V_N, and Ta³⁺ defects, the intermediate states mainly consist of Ta 5*d* and N 2*p* orbitals (Supplementary Fig. 4a). Upon doping La into Ta₃N₅ lattice, the intermediate states mostly originate from Ta 5*d*, N 2*p* orbitals, and a small amount of La 5*d* orbital (Supplementary Fig. 4b). The calculated total density of states for undoped Ta₃N₅ and La-doped Ta₃N₅ are plotted in Fig. 1e. The undoped Ta₃N₅ shows a high density of intermediate states, which is consistent with our previous results²³. In contrast, the density of intermediate states is significantly reduced in La-doped Ta₃N₅."

Reference

23. Fu, J., *et al.* Identifying performance-limiting deep traps in Ta₃N₅ for solar water splitting. *ACS Catal.* **10**, 10316-10324 (2020).

Comment 5: *On line 117 you write, "Quantitative XPS analyses show that the La/Ta ratio in the La-doped Ta₃N₅ thin film is 2.97%." Can the authors comment on the synthesis and if this was the expected surface doping level.*

Response 5: The La-doped oxide precursor films were prepared by dual-source electron beam physical vapor deposition using the La₂O₃ (99.99% in purity) and Ta₂O₅ (99.99% in purity) as the evaporation sources. The deposition rates of Ta₂O₅ and La₂O₃ were monitored by quartz crystal microbalance to control the thickness and La content of the deposited films. For 3% La-doped films, the deposition rates of Ta₂O₅ and La₂O₃ were controlled at approximately 5.0 and 0.2 Å/s, respectively. The atomic ratio of La/Ta is estimated to be 3.5% according to the following equation:

$$\frac{n_{\text{La}}}{n_{\text{Ta}}} = \frac{\rho_{\text{La}_2\text{O}_3} A_{\text{La}_2\text{O}_3} M_{\text{Ta}_2\text{O}_5}}{\rho_{\text{Ta}_2\text{O}_5} A_{\text{Ta}_2\text{O}_5} M_{\text{La}_2\text{O}_3}}$$

where ρ is the density of source material, A is the deposition rate, and M is the molecular weight. Considering the deposited oxide materials may not be stoichiometric, the actual atom ratios of La/Ta in the thin films were measured using EDS (**Supplementary Fig. 11**). Moreover, quantitative XPS analysis shows that the La/Ta concentration in the 3% La-doped Ta₃N₅ thin film is 2.97%, consistent with the EDS result (2.77%).

Comment 6: *In Figure S7, you write in the caption for c and d that the applied potential was 1.0 V vs RHE; however, in the figures themselves you have written 1.23 V vs RHE. Please clarify. Also, you state that samples were measured under AM 1.5G radiation. Although it is obvious, please make it clear to the reader that this was using a solar simulated light source.*

Response 6: We thank the reviewer for his/her careful reading of the manuscript. We have corrected the errors in the Supplementary Information according to the reviewer's remarks: "1.0 V vs RHE"

in the caption of **Supplementary Fig. 12** was revised as "1.23 V vs RHE", and "AM 1.5G" was revised as "AM 1.5G simulated sunlight" in the caption of **Supplementary Figs. 9, 12, and 13**.

Comment 7: On line 171 you write, "The significantly reduced grain size leads to severe carrier recombination at the grain boundaries". This statement is too conclusive. Instead you should infer that this may have been caused by the decrease in average crystal size displayed in samples with higher levels of La doping.

Response 7: We thank the reviewer for pointing out the inappropriate statement. We have revised the manuscript according to the reviewer's suggestions. The detailed revisions are listed below:

On lines 186-192: "The grain size of the thin films can potentially impact charge transport properties through trap states at grain boundaries that act as recombination centers³⁶. The decrease in average grain sizes leads to the formation of a large number of grain boundaries in the thicker films, resulting in carrier recombination at the grain boundaries. Therefore, the poor PEC performance in the thicker (500 and 700 nm) 3% La-doped Ta₃N₅ photoanodes (Supplementary Fig. 9) may have been caused by the decrease in average grain size with a high La doping level."

Reference

36. Pinaud, B. A., Vesborg, P. C. K. & Jaramillo, T. F. Effect of Film Morphology and Thickness on Charge Transport in Ta₃N₅/Ta Photoanodes for Solar Water Splitting. *J. Phys. Chem. C* **116**, 15918-15924 (2012).

Comment 8: The data for undoped Ta₃N₅ at various thicknesses shown in Figure S7 is not discussed in the main text. Please provide reasoning to the reader for why the 700 nm thick Ta₃N₅ shows a higher photocurrent density than that of any La-doped Ta₃N₅ photoanode produced herein.

Response 8: We are sorry that we did not discuss the data for undoped Ta₃N₅ at various thicknesses shown in **Supplementary Fig. 7** in the previous manuscript. Since the undoped samples are used as reference samples, they are not discussed in detail in the main text. We have added more discussion on **Page 8** of our revised manuscript.

The thicker (500 and 700 nm) Ta₃N₅ films do show a higher photocurrent density than that of 3% La-doped Ta₃N₅ photoanode with the same thickness produced. Although La doping improves the light absorption and reduces the Ta³⁺-related trap density in Ta₃N₅, it deteriorates the carrier transport due to significantly reduced grain size and increased effective masses of electrons and holes. Especially in thicker films, the relatively small grain sizes produce a large number of grain

boundaries acting as the recombination centers, which are detrimental to photogenerated carrier transport. Therefore, the poor PEC performance in thicker (500 and 700 nm) 3% La-doped Ta₃N₅ photoanodes (**Supplementary Fig. 9**) may have been caused by the decrease in average crystal size with a high La doping level. By reducing the La doping concentration to 2% in the 700 nm film to reduce the side effect caused by the grain size reduction, it is possible to improve the photocurrent density to 9.23 mA cm⁻² at 1.23 V versus RHE (**Supplementary Fig. 13**). In the revised manuscript, we have added some descriptions according to the reviewer's suggestions.

The related information has now been included/updated on page 8 of the main text.

On lines 155-160: "For undoped Ta₃N₅ photoanodes, the light absorption and photocarrier utilization are enhanced with increasing film thickness up to 700 nm, resulting in a continuous increase in photocurrent (Supplementary Fig. 9a and 9c). For La-doped Ta₃N₅ photoanodes, the photocurrent was improved to a lesser extent at a thickness of 300 nm, and even decreased for thicker (500 and 700 nm) films (Supplementary Fig. 9b and 9d)."

Comment 9: On line 184, you write, "The above results show that although La doping improves light absorption..." Again, the statement "show" is too strong. Rather, your data "indicates" this.

Response 9: Thank you very much for your correction. We have changed "show" to "indicate" on **line 203** of the revised manuscript.

Comment 10: I recommend that authors introduce a table that summarises the PEC data of all samples produced herein. This can go in the SI. One way of presenting this data could be the photocurrent density at a specific RHE. The doping level can be on the top row and the columns on the left can be the film thickness. You can also do this for maximum "ABPE" (or preferably STH, as I state in point 1) and the photocatalytic onset potential if you wish also.

Response 10: Thank you very much for this valuable suggestion. We have summarized the PEC performance data for all samples prepared in this study in **Supplementary Table 1**, including the La-doping level, the film thickness, the photocurrent density at 1.23 V RHE, the onset potential, and the HC-STH.

Supplementary Table 1 | The detailed parameters of the J-V curves for Ta₃N₅-based photoanodes in this study. The onset potential is defined as the potential at a photocurrent density of 0.1 mA cm⁻². For heterogeneous doped Ta₃N₅ photoanodes, the steady-state photocurrent at low bias conditions was measured to determine the onset potential.

Undoped and La-doped Ta₃N₅ photoanodes				
Doping level	Thickness (nm)	Onset potential (V versus RHE)	Photocurrent at 1.23 V versus RHE (mA cm ⁻²)	HC-STH (%)
undoped	100	0.63	1.51	0.30
	300	0.58	3.68	0.93
	500	0.55	5.32	1.06
	700	0.50	6.18	1.37
2%	100	0.59	3.52	0.67
	700	0.51	9.23	2.05
3%	100	0.58	4.40	0.80
	300	0.57	5.04	0.87
	500	0.58	4.02	0.60
	700	0.59	3.77	0.52
4%	100	0.64	3.41	0.55
Heterogeneous doped Ta₃N₅ photoanodes				
Gradient-Mg:Ta ₃ N ₅		0.39	8.51	3.30
Gradient-Mg:Ta ₃ N ₅ /Ta ₃ N ₅		0.42	8.17	3.00
Gradient-Mg:Ta ₃ N ₅ /La:Ta ₃ N ₅		0.39	10.06	4.07

Comment 11: On line 209 you write, “However, the La dopants in the surface layer cannot be directly detected by EDS mapping due to its low concentration.” Also, Figure 4a implies that there is no Mg doping at the surface of the material, and that is solely lies in the material bulk. To clarify what the surface and bulk constituents are in your material I strongly recommend XPS analysis of the surface and bulk (through sputtering) of your La and Mg co-doped Ta₃N₅ material. I recommend you investigate La, Mg, Ta, N and O content.

Response 11: We thank the reviewer for raising this important question. During the high-temperature nitridation process, it is inevitable that Mg dopants may diffuse into the surface layer. Following the reviewer’s suggestion, we have conducted XPS depth profile analyses of the prepared gradient-Mg:Ta₃N₅/La:Ta₃N₅ film on Nb substrate to clarify its surface and bulk constituents (**Supplementary Fig. 16**). The XPS depth profiles reveal the presence of low concentration of La dopants in the surface layer and the gradient distribution of Mg dopants in the inner layer of the film. The presence of Mg in the surface La:Ta₃N₅ layer due to the diffusion of Mg dopants during

high-temperature nitridation is also observed. Nevertheless, the morphological (compact surface layer with small grain size in **Supplementary Figs. 14a** and **18a,b**) and photoelectrochemical (enhanced solar photocurrent in the range of 480-590 nm in **Supplementary Fig. 23**) properties indicate that the predominant doping effect in the surface layer comes from La, instead of Mg. Therefore, in order not to overcomplicate the system, we named the surface layer La:Ta₃N₅, instead of La-Mg codoped Ta₃N₅. However, we have pointed out in the revised manuscript the diffusion of Mg into the surface layer.

The related information has now been provided on page 12 of the main text, and in **Supplementary Fig. 16** of the Supplementary Information.

On line 234-237: "The XPS depth profiles further confirm the doping of La in the surface layer (Supplementary Fig. 16). Furthermore, it is revealed that besides the gradient distribution of Mg dopants in the Mg:Ta₃N₅ layer, they may diffuse into the surface layer during the high-temperature nitridation process."

Supplementary Fig. 16 | XPS depth profile for gradient-Mg:Ta₃N₅/La:Ta₃N₅ film on Nb substrate. a-f, Ta 4f, Mg 1s, La 3d, N 1s, O 1s, and Nb 3d peaks as a function of etching time, respectively. g, XPS depth profile analyses of elemental concentrations. h-i, The change of atom

ratios of Mg/Ta and La/Ta in the film with etching time. In the high binding energy region of Mg 1s, surface damage due to ion etching results in a decrease in peak intensity accompanied by an increase in the background level on the high binding energy side (*Ref. 2*). In order to make a more reliable quantitative analysis, a physically more meaningful Tougaard-type background subtraction method was employed for peak fitting of the XPS peaks of Mg 1s. The Tougaard-type background takes into account the inelastic scattering of electrons and has been widely used in quantitative XPS analysis (*Ref. 3*).

Supplementary References

2. Greczynski, G. & Hultman, L. X-ray photoelectron spectroscopy: Towards reliable binding energy referencing. *Prog. Mater. Sci.* **107**, 100591 (2020).
3. Tougaard S. Practical guide to the use of backgrounds in quantitative XPS. *J. Vac. Sci. Technol. A* **39**, 011201 (2021)

Comment 12: *Figure 16d should also be produced for the Mg-gradient doped Ta₃N₅ samples shown in Figure 16c, for comparison. Personally, it would be better if Figure 16b and c were merged. And similarly, when constructing this new figure I have now requested, it be merged with Figure 16d also.*

Response 12: We thank the reviewer for this helpful suggestion and have replotted this figure (**Supplementary Fig. 21**) accordingly.

Supplementary Fig. 21 | Reproducibility of the PEC performance for gradient-Mg:Ta₃N₅/La:Ta₃N₅ photoanodes. All samples were modified with NiCoFe-B₁ cocatalyst and tested under AM 1.5G simulated sunlight in 1 M KOH. **a**, *J-V* curves for a batch of 25 gradient-Mg:Ta₃N₅/La:Ta₃N₅ photoanodes. **b**, Statistics of the photocurrents for the gradient-Mg:Ta₃N₅/La:Ta₃N₅ photoanodes in (a) and gradient-Mg:Ta₃N₅ photoanodes (data extracted from *Ref. 1*) at 1.23 V versus RHE. **c**, Statistics of the HC-STHs for the gradient-Mg:Ta₃N₅/La:Ta₃N₅ photoanodes in (a) and gradient-Mg:Ta₃N₅ photoanodes (data extracted from *Ref. 1*).

Supplementary Reference

1. Xiao, Y. *et al.* Band structure engineering and defect control of Ta₃N₅ for efficient photoelectrochemical water oxidation. *Nat. Catal.* **3**, 932-940 (2020).

Comment 13: On line 262 you write “1.0 V versus RHE” but again the figure itself writes 1.23 V vs RHE. Please clarify.

Response 13: Thank the reviewer very much for pointing out this error. We have checked our manuscript carefully and corrected the mistakes in the revised manuscript.

Reviewer #2:

General comment 1: This paper uses a heterogeneous doping strategy to improve the photoabsorption properties of Ta_3N_5 at photoanode surface ($La:Ta_3N_5$) while preserving the efficient carrier transport properties of the $Mg:Ta_3N_5$ thin film, in doing so achieving a very high ABPE efficiency of 4.07%, a record for Ta_3N_5 based photoanodes. The mechanism of increased photoabsorption through La doping is attributed to reduced optical anisotropy, or in other words, slightly disrupting the Ta_3N_5 lattice. The La^{3+} doping is also claimed to improve the charge transfer properties of Ta_3N_5 films, notably only compared to bare Ta_3N_5 film rather than other doped Ta_3N_5 films like $Mg:Ta_3N_5$, by the introduction of O_N donors and the suppression of V_N and Ta^{3+} defects when the concentration of La^{3+} is not too high (~3%), otherwise the charge transfer properties suffer when the lattice structure of Ta_3N_5 is greatly distorted. The charge transfer performance is further aided by using $Mg:Ta_3N_5$ on the back of $La:Ta_3N_5$.

This paper introduces a novel way of reducing optical anisotropy by “lattice engineering” via doping, which is industrially relatively easy to implement, and shows up to 100% more photoabsorption at wavelengths close to bandgap, and consistently improved photoabsorption at photoenergies greater than bandgap. The performance of the $La:Ta_3N_5$ film developed to fulfill such goal was thoroughly measured, including honestly admitting that the charge transfer properties of such film suffers from disturbed lattice structure, and the charge transfer properties of $La:Ta_3N_5$ is considerably worse than that of $Mg:Ta_3N_5$, or even Ta_3N_5 in some cases. This paradox of light absorption and charge transfer calls for optimization of La dopant concentration and $La:Ta_3N_5$ film thickness, which the authors carried out systematically. Thus, engineering wise, the authors explored a lot of parameter space and provided enough information to validate the effectiveness of their photoanode.

General Response: We genuinely appreciate the reviewer’s positive evaluation of our work. The constructive comments made by the reviewer have helped to further improve the quality of our work. We have carefully revised the manuscript according to the comments and the point-by-point responses are provided below.

Comment 1: However, despite showing the XPS survey spectra of $La:Ta_3N_5$ and Ta_3N_5 , the authors did not explicitly specify why La is the go to dopant for tuning optical anisotropy of Ta_3N_5 , and what are the shortcomings of other dopant candidates.

Response 1: Because the conduction band (CB) of Ta_3N_5 consists of the unoccupied Ta 5d orbitals (Murthy *et al.*, *Chem. Sci.*, **2019**, *10*, 5353-5362), doping with transition metal with different d-

orbital energy can tune the CB structure in theory. In general, dopants with lower d -orbital energy than that of Ta can shift down the CB edge, resulting in a smaller bandgap. For example, the W-doped Ta₃N₅ synthesized by Grigorescu *et al.* has a band gap of 1.75 eV [Grigorescu *et al.*, *Electrochem. Commun.*, **2015**, *51*, 85-88]. But the narrowing of the bandgap may negatively affect the photogenerated voltage of Ta₃N₅-based photoanodes. This can be avoided with a dopant having higher d -orbital energy, since the narrowing of the bandgap is less in this case [Modak and Ghosh, *Sol. Energy Mater. Sol. Cells*, **2017**, *159*, 590-598].

In this work, the La dopant has a similar valence electron configuration to that of Ta (La: $5d^16s^2$, Ta: $5d^36s^2$) and a higher $5d$ orbital energy due to the special $4f$ electron configuration ($4f^0$) (<https://www.tutorsglobe.com/homework-help/chemistry/nature-and-chemistry-of-transition-elements-78754.aspx>). When Ta is partially substituted by La, the hybridization of La $5d$ and Ta $5d$ orbitals may lead to more delocalized orbital distribution in the CB [Kamarulzaman *et al.* *Nanoscale Res. Lett.* **2015**, *10*, 346]. Indeed, the DFT calculations show that La doping leads to an increased density of states and a more delocalized orbital distribution near the CBM and VBM (**Supplementary Fig. 2 and Fig. 1b**), which may contribute to the enhanced light absorption in the range of 480-590 nm. Moreover, La doping only narrows the optical bandgap of Ta₃N₅ by 0.02 eV (**Supplementary Fig. 1b**). However, as we discussed in the manuscript, La doping also brings some negative effects, such as poorer carrier transport due to reduced grain size and increased effective mass of electrons and holes. Alternative dopants that could enhance optical absorption without hindering carrier transport are still under pursuit. A promising candidate may be Hf⁴⁺ (71 pm) with an ionic radius similar to that of the six-coordinated Ta⁵⁺ (64 pm). The valence electron configurations of Hf and Ta are $5d^26s^2$ and $5d^36s^2$. Through DFT calculations, Wang *et al.* found that the dopants Hf were theoretically able to maintain the carrier mobility of Ta₃N₅ [Wang *et al.*, *Appl. Catal. B Environ.*, **2019**, *244*, 502-510].

The related information has now been provided on page 5 of the main text.

On line 77-83: "The conduction band (CB) of Ta₃N₅ mainly consists of the unoccupied Ta $5d$ orbitals, while the valence band (VB) is mainly composed of N $2p$ orbitals²⁰. La has a similar valence electron configuration to that of Ta (La: $5d^16s^2$, Ta: $5d^36s^2$) and a higher $5d$ orbital energy due to the special $4f$ electron configuration ($4f^0$). When Ta is partially substituted by La in Ta₃N₅, the hybridization of La $5d$ and Ta $5d$ orbitals may lead to more delocalized orbital distribution in the CB, which may contribute to the enhanced light absorption."

Reference

20. Nurlaela, E., et al. Combined experimental and theoretical assessments of the lattice dynamics and optoelectronics of TaON and Ta₃N₅. *J. Solid State Chem.* **229**, 219-227 (2015).

Comment 2: From our understanding, La doping have been used in La:STO systems for decades for its electrochemical properties, i.e. reduce electron concentration, and La:STO coating have already been introduced to the photocatalysis community (<https://doi.org/10.1016/j.solmat.2020.110428>). This begs the question, what are the significant differences, electrochemical mechanism wise, between La doping in La:STO and La:Ta₃N₅?

Response 2: SrTiO₃, as a perovskite structure (molecular formula ABO₃), has two distinct cationic sites in its structure. The B-site cation forms a BO₆ octahedron with the anions in six-fold coordination, while the A cation is surrounded by the octahedron in twelve-fold coordination. The A atoms are generally larger than the B atoms. According to the Goldschmidt tolerance factor, an indicator of perovskite structural stability [Goldschmidt, *Naturwissenschaften*, **1926**, *14*, 477-485], the ionic radii of La³⁺ (103 pm for six-coordinate and 136 pm for twelve-coordinate) is more close to twelve-coordinate Sr²⁺ (146 pm) than to six-coordinate Ti⁴⁺ (61 pm). Therefore, La³⁺ tends to substitute for Sr²⁺ rather than Ti⁴⁺ [Patil *et al.*, *J. Environ. Chem. Eng.*, **2020**, *8*,103791]. The valence band (VB) and the conduction band (CB) of SrTiO₃ are dominantly comprised of occupied O 2p states and unoccupied Ti 3d states, respectively [Qureshi *et al.*, *J. Catal.* 2019, *376*, 180-190]. Theoretical study by Li *et al.* found that the substitution of La for Sr does not change the structure of the Ti-O octahedron, so it has less effect on light absorption [Li *et al.*, *Mater. Sci. Eng. B-Adv.*, **2010**, *172*, 136-141]. However, La (5d¹6s²) has more valence electrons than Sr (5s²), and the La dopant atom doped at the A-site releases more electrons into the SrTiO₃ lattice than the Sr atom do, so La dopant atoms are donor impurities, resulting in increased the electron concentration [Miyachi *et al.*, *Langmuir*, **2004**, *20*, 232-236].

In the study of Lucas *et al.* referred by the reviewer, La³⁺ is doped into Sr²⁺ sites, which suppresses the deep trapping states related to formed SrO species, resulting in enhanced water-splitting performance of the photoanodes [Lucas *et al.*, *Sol. Energy Mater. Sol. Cells*, **2020**, *208*, 110428]. Others claim that La doping greatly enhances the photocatalytic activity of SrTiO₃ due to formation of particles with smaller size and good crystallinity, which reduces the charge carrier recombination rate [Li *et al.*, *Mater. Sci. Eng., B*, **2010**, *172*, 136-141]. Besides, some researchers have found that La dopants can balance the charge of the SrTiO₃ system, especially for co-doping strategies [Patil *et al.*, *J. Environ. Chem. Eng.*, **2020**, *8*,103791]. The effective role of La as a charge-compensating

element has been involved in many cases, such as N-doped [Miyachi *et al.*, *Langmuir*, **2004**, *20*, 232-236], Cr-doped [Tonda *et al.*, *Phys. Chem. Chem. Phys.*, **2014**, *16*, 23819-23828], Rh-doped [Wang *et al.*, *Chem. Mater.*, **2014**, *26*, 4144-4150], Al-doped SrTiO₃ [Qin *et al.*, *ACS Catal.*, **2021**, *11*, 11429-11439], etc.

In our study, La replacing Ta improves light absorption near the absorption edge, probably due to the increased density of states and more delocalized orbital distribution near the CBM and VBM after La doping. In addition, La doping introduces more O_N donors into the Ta₃N₅ lattice, which leads to an increase in the carrier concentration. Therefore, although La doping improves the photocatalytic/photoelectrochemical water-splitting activity of SrTiO₃ and Ta₃N₅, the fundamental mechanism may be different.

Comment 3: *Is La doping a universal solution to increase photoabsorption performance across all common photoanode materials?*

Response 3: As we mentioned in the above comments, replacing Ta by La in Ta₃N₅ improves light absorption, probably attributed to the increased density of states and more delocalized orbital distribution near the CBM and VBM. This mechanism mainly involves the inter-hybridization of *d* orbitals with different energies, resulting in a change in the CB structure [Modak and Ghosh, *Sol. Energy Mater. Sol. Cells*, **2017**, *159*, 590-598]. Therefore, a similar effect may be brought about if La is doped into semiconductor materials whose CBM are mainly composed of *d* orbitals. Notably, the target atoms substituted by La are the major contributors to the CBM. The effectiveness of La doping to improve light absorption has also been confirmed in other semiconductor materials, such as ZnO [Goel *et al. Physica E*, 2017, *91*, 72-81], TiO₂ [Zhao and Liu, *J. Phys. D Appl. Phys.*, **2008**, *41*, 085417], BiVO₄ [Min *et al.*, *J Rare Earth*, **2013**, *31*, 878-884], and Ag₃PO₄ [Xie and Wang, *J. Colloid Interface Sci.*, 2014, *430*, 1-5], etc. Nevertheless, it should be pointed out that the mechanisms claimed by these authors are slightly different from ours, and, to the best of our knowledge, the detailed mechanism by which La doping enhances light absorption remains elusive at this moment. Here we only propose a hypothesis based on our experimental data and try not to oversell our findings.

Comment 4: *Additionally, the electrochemical performance benchmark for La:Ta₃N₅ to compare to should not only be Ta₃N₅, but also Mg:Ta₃N₅, so the XPS data of Mg:Ta₃N₅ should be added as well, along with a small discussion of light absorbing properties of Mg:Ta₃N₅.*

Response 4: Thank the reviewer very much for his/her helpful suggestion. The role of Mg doping in improving the PEC water splitting for Ta₃N₅ photoanode has been discussed in our previous work

[Xiao *et al*, *Nat. Catal.*, **2020**, *3*, 932-940], and the main reason has been ascribed to that Mg doping efficiently reduces deep traps created by nitrogen vacancies and increase shallow donors generated by oxygen impurities in Ta₃N₅. Here, as per the reviewer's comment, we have compared the light-absorbing properties and XPS data of La-doped and Mg-doped Ta₃N₅ film with a thickness of 100 nm (**Supplementary Fig. 1** and **Supplementary Fig. 6**). Compared with the undoped Ta₃N₅ film, the above-bandgap light absorption is weakened in Mg-doped sample, while the La-doped sample is notably enhanced in the range of 480-590 nm (**Supplementary Fig. 1a**). Moreover, Mg doping causes an increase in the optical bandgap, while La doping narrows the bandgap of Ta₃N₅ films (**Supplementary Fig. 1b**). On the other hand, the XPS results show that Mg doping increases both oxygen and nitrogen contents in Ta₃N₅ films (**Supplementary Fig. 6**), while La doping has a greater effect on the increase in oxygen content. In our revised manuscript, we have added the XPS data and UV-vis absorption spectra of Mg-doped Ta₃N₅ film with relevant descriptions and references. We thank the reviewer for the helpful comment improving the quality of our manuscript.

We have included the related information in Supplementary Figs. 1 and 6, and in the main text on page 7.

On line 137-139: "For Mg-doped Ta₃N₅ film, both nitrogen and oxygen contents are increased, the N/Ta atom ratio is 1.62 and the O/Ta atom ratio is 0.30 (Supplementary Fig. 6), which is consistent with our previous results²⁴."

References

24. Xiao, Y., et al. Band structure engineering and defect control of Ta₃N₅ for efficient photoelectrochemical water oxidation. *Nat. Catal.* **3**, 932-940 (2020).

Supplementary Fig. 1 | Optical absorption properties of undoped, La-doped and Mg-doped Ta₃N₅ films on quartz substrates. a, UV-vis absorption spectra. The thickness of all films is about 100 nm. The Mg/Ta concentration in the Mg-doped Ta₃N₅ thin film was 13.5% estimated from XPS

results (Supplementary Fig. 6). Compared with the undoped Ta₃N₅ film, the above-bandgap light absorption is weakened in the Mg-doped sample, while that of the La-doped sample is notably enhanced in the range of 480-590 nm. **b**, Tauc plots of UV-vis absorption spectra. α , absorption coefficient; h , Planck's constant; ν , photon's frequency. The dotted lines show the extrapolation of the linear portion of the absorption edges. Mg doping causes an increase in the optical bandgap, while La doping narrows the bandgap of Ta₃N₅ films.

Supplementary Fig. 6 | XPS core-level spectra of Mg-doped Ta₃N₅. **a**, Mg 1s peak. **b**, Ta 4f peak. **c**, O 1s peak. **d**, N 1s peak. Quantitative XPS analysis showed that the Mg/Ta concentration was approximately 13.5%, and the N/Ta and O/Ta atomic ratios were 1.62 and 0.30, respectively. These results are consistent with our previous studies, which suggested that Mg doping can effectively reduce deep traps created by nitrogen vacancies and increase shallow donors generated by oxygen impurities in Ta₃N₅ (Ref. 1).

Supplementary References

1. Xiao, Y. *et al.* Band structure engineering and defect control of Ta₃N₅ for efficient photoelectrochemical water oxidation. *Nat. Catal.* **3**, 932-940 (2020).

Comment 5: In addition to the properties of the La:Ta₃N₅ coating itself, when Ni based Oxygen evolving catalyst is conformally introduced to the coating, the authors should give a brief discussion about the interface between catalyst and the coating, i.e. the transportation properties of charge carriers.

Response 5: We thank the reviewer for this valuable suggestion. The transport properties of charge carriers at the interface between NiCoFe-Bi cocatalyst and coating have indeed been neglected in the previous manuscript. We have added the photoelectrochemical impedance spectroscopy analysis of the photoelectrodes and given a brief discussion in the revised manuscript. Impedance spectroscopy analysis reveals that the charge transfer resistance across the photoanode/electrolyte interface was significantly reduced after NiCoFe-Bi cocatalyst modification (**Supplementary Fig. 19** and **Supplementary Table 4**). This indicates that the deposition of NiCoFe-Bi cocatalyst on the photoanode surface improves the charge separation and transfer conditions [Fu *et al.*, *Adv. Funct. Mater.*, **2018**, 28, 1706785; Wei *et al.*, *Small*, **2021**, 17, 2100084].

The related information has now been provided in the main text on page 14, and in Supplementary Fig. 19 and Supplementary Table 4 of the Supplementary Information.

On lines 274-279: "Moreover, photoelectrochemical impedance spectroscopy analysis reveals that the charge transfer resistance across the photoanode/electrolyte interface is significantly reduced after NiCoFe-Bi cocatalyst modification (Supplementary Fig. 19 and Supplementary Table 4). This indicates that the cocatalyst modification is necessary to improve the charge transfer kinetics across the photoanode/electrolyte interface^{40, 41}."

References

40. Zhang, K., et al. Black phosphorene as a hole extraction layer boosting solar water splitting of oxygen evolution catalysts. *Nat. Commun.* 10, 2001 (2019).
41. Wei, S., et al. Selective cocatalyst deposition on ZnTiO_{3-x}N_y hollow nanospheres with efficient charge separation for solar-driven overall water splitting. *Small* 17, 2100084 (2021).

Supplementary Fig. 19 | Photoelectrochemical impedance spectroscopy (PEIS) for the gradient-Mg:Ta₃N₅/La:Ta₃N₅ photoanode with/without NiCoFe-B₁ cocatalyst modification.

The PEIS data were measured in 1 M KOH electrolyte at 1.0 V versus RHE under AM 1.5G simulated sunlight. Green lines denote the fitting of the PEIS data. The inset shows a two-RC-unit equivalent circuit used to fit the Nyquist plot of the PEIS, which consists of three resistances and two capacitors: a series resistance (R_s) of the electrolyte, external contact, and conductive substrate layer, a bulk charge transport resistance (R_{trap}), a semiconductor/electrolyte charge transfer resistance (R_{ct}), a bulk capacitor of space charge region (C_{bulk}), and a surface states capacitor (C_{ss}). The photoanode without cocatalyst modification showed a larger semicircle diameter, while the photoanode with the cocatalyst modification showed a smaller semicircle diameter, indicating that the charge transfer resistance of the photoanode was reduced after the NiCoFe-B₁ cocatalyst modification. The fitted values of R_s , R_{trap} and R_{ct} from the equivalent circuit are displayed in Supplementary Table 4.

Supplementary Table 4 | Fitted values of R_s , R_{trap} , and R_{ct} of the gradient-Mg:Ta₃N₅/La:Ta₃N₅ photoanode with/without NiCoFe-B₁ cocatalyst modification.

Photoanodes	R_s (Ohm)	R_{trap} (Ohm)	R_{ct} (Ohm)
Gradient-Mg:Ta ₃ N ₅ /La:Ta ₃ N ₅	2.0	3.2	913.9
Gradient-Mg:Ta ₃ N ₅ /La:Ta ₃ N ₅ /NiCoFe-B ₁	1.9	2.1	268.3

Reviewer #3:

General comment: As one of the most promising photoanode materials for solar-driven photoelectrochemical (PEC) water splitting, tantalum nitride (Ta_3N_5) has received tremendous research attention in recent years. A steady improvement in the applied bias photon to current efficiency (ABPE) has been achieved through different strategies such as nanostructure engineering, defects engineering, and interface engineering. Here, Li et al. report a heterogeneous doping strategy to decouple the light absorption and carrier transport in Ta_3N_5 thin-film photoanode. A maximum ABPE of 4.07% is obtained for PEC water splitting, which to the best of my knowledge, is the highest among reported values for Ta_3N_5 photoanodes. The result is significant and very exciting, especially the effect of La doping on alleviating anisotropic optical absorption in Ta_3N_5 , which was neglected in previous studies of Ta_3N_5 photoanodes. The proposed heterogeneous doping strategy could also serve as a guideline for improving the light conversion efficiency of other semiconductor thin films by breaking performance trade-offs between light absorption and carrier transport. Overall, this study is very interesting and the paper is well-articulated.

General Response: We greatly appreciate the reviewer for the very positive evaluation of our work. We further would like to thank the reviewer for the constructive comments which helped to further improve the quality of the paper. In the revised manuscript, we have considered all of the points that were raised. We sincerely hope that the details provided below will address the reviewer's concerns.

The paper can be published after addressing the following minor remarks:

Comment 1: If I understood correctly, the term “pristine Ta_3N_5 ” refers to an ideal Ta_3N_5 crystal without any defects while “undoped Ta_3N_5 ” refers to Ta_3N_5 with intrinsic defects such as nitrogen vacancies and reduced Ta^{3+} species. The correct use of the terms should be checked throughout the manuscript to avoid confusion.

Response 1: Thank you very much for your kind suggestions. We have checked our manuscript carefully and modified the mistakes in the revised manuscript.

Comment 2: For the valence band positions of La-doped Ta_3N_5 film, why do the values obtained by XPS valence band spectra (Fig. S6) and USP spectra (Fig. 4c) differ significantly?

Response 2: To the best of our knowledge, the UPS uses He I state with the photon energy of 21.22eV to produce ultraviolet light and has resolutions up to 0.01 eV (~10 meV). In contrast, XPS uses Al Ka line with an X-ray energy of 1486.6 eV and has resolutions ranging from 0.3 eV to 0.7 eV. The low accuracy of XPS leads to a large error in the determined valence band position.

Moreover, the improved resolution of UPS compared with XPS suggests that UPS can detect more refined structures of valence band spectra.

Comment 3: The pH condition for the potential scale of the normal hydrogen electrode in Figure 4d should be given.

Response 3: We thank the reviewer for pointing out the missing part in the manuscript. We have fixed Fig. 4d, which now has a new label of “Potential vs. NHE at pH = 0 (V)”.

Comment 4: Fig. 5a shows that the gradient-Mg:Ta₃N₅/La:Ta₃N₅ photoanode has a higher photocurrent than that of the gradient-Mg:Ta₃N₅ photoanode. Since the two samples have different thicknesses, it is better to conduct a control experiment to compare samples with the same Ta₃N₅ film thickness. For instance, the surface La-doped Ta₃N₅ layer can be replaced with an undoped Ta₃N₅ layer having the same thickness on the gradient Mg-doped samples for comparison.

Response 4: Thank the reviewer very much for raising this point. Following the reviewer’s suggestion, we have tested the solar-driven PEC water oxidation properties of the gradient-Mg:Ta₃N₅/Ta₃N₅ modified with NiCoFe-B_i cocatalysts as a reference experiment. The data is provided as **Supplementary Fig. 20** in the revised Supplementary Information. The surface undoped Ta₃N₅ layer in the gradient-Mg:Ta₃N₅/Ta₃N₅ film has the same thickness as the La-doped Ta₃N₅ layer in the gradient-Mg:Ta₃N₅/La:Ta₃N₅ film. The gradient-Mg:Ta₃N₅/Ta₃N₅ photoanode showed a decreased photocurrent density of 8.2 mA cm⁻² at 1.23 V versus RHE, indicating that the increased photocurrent of gradient-Mg:Ta₃N₅/La:Ta₃N₅, compared with gradient-Mg:Ta₃N₅, was not due to an increase in the thickness of the photoanode. A steady photocurrent density of ~20 μA cm⁻² was generated at 0.42 V versus RHE, revealing that the introduction of an undoped layer of Ta₃N₅ resulted in a positive shift of onset potential for the gradient Mg:Ta₃N₅/Ta₃N₅ photoanode.

The related information has now been provided in the main text on pages 14-15, and in Supplementary Fig. 20 of the Supplementary Information.

On lines 281-287: "To exclude potential influence of the film thickness, the PEC properties of the gradient-Mg:Ta₃N₅/Ta₃N₅ photoanode are also measured for comparison, as shown in **Supplementary Fig. 20**. The gradient-Mg:Ta₃N₅ photoanode achieves a photocurrent density of 8.51 mA cm⁻² at 1.23 V versus RHE, which is consistent with our previous report²⁴. The gradient-Mg:Ta₃N₅/Ta₃N₅ photoanode showed a decreased photocurrent density of 8.2 mA cm⁻² at 1.23 V versus RHE."

Reference

24. Xiao, Y., et al. Band structure engineering and defect control of Ta₃N₅ for efficient photoelectrochemical water oxidation. *Nat. Catal.* **3**, 932-940 (2020).

Supplementary Fig. 20 | Solar-driven PEC water oxidation properties of gradient-Mg:Ta₃N₅/Ta₃N₅ modified with NiCoFe-B₁ cocatalysts. **a**, *J-V* curves under AM 1.5G simulated sunlight in 1 M KOH. **b**, Steady-state photocurrent of gradient-Mg:Ta₃N₅/Ta₃N₅ photoanode under low applied potentials. **c**, Steady-state photocurrent at an applied potential of 1.0 V versus RHE for 120 min. **d**, HC-STH of the photoanodes calculated from *J-V* curves in (a). The surface undoped Ta₃N₅ layer in the gradient-Mg:Ta₃N₅/Ta₃N₅ film has the same thickness as the La-doped Ta₃N₅ layer in the gradient-Mg:Ta₃N₅/La:Ta₃N₅ film. The gradient-Mg:Ta₃N₅/Ta₃N₅ photoanode showed a decreased photocurrent density of 8.2 mA cm⁻² at 1.23 V versus RHE, indicating that the increased photocurrent of gradient-Mg:Ta₃N₅/La:Ta₃N₅, compared with gradient-Mg:Ta₃N₅, was not due to an increase in the thickness of the photoanode. A steady photocurrent density of ~20 μA cm⁻² was generated at 0.42 V versus RHE, revealing that the introduction of an undoped layer of Ta₃N₅ resulted in a positive shift of onset potential for the gradient Mg:Ta₃N₅/Ta₃N₅ photoanode.

Comment 5: The abbreviation “FE” in line 592 was not defined before use.

Response 5: Thank the reviewer very much for pointing out this error. The abbreviation "FE" stands for Faraday Efficiency. We have corrected this in our revised manuscript.

REVIEWERS' COMMENTS

Reviewer #1 (Remarks to the Author):

The authors have adequately addressed all of the minor concerns that I had with the manuscript, including conducting additional XPS depth profile experiments. I am satisfied with their changes and not recommend their work for publication.

Reviewer #2 (Remarks to the Author):

The authors have addressed my concerns of the original manuscript clearly and professionally. I recommend the publication of this paper in your journal at this point.

Reviewer #3 (Remarks to the Author):

This manuscript has been properly revised and can be published as it is.

Point-by-Point Responses to Reviewers' Comments

Manuscript number: NCOMMS-22-37115A

Manuscript type: Research Article

Title: "Decoupling light absorption and carrier transport via heterogeneous doping in Ta₃N₅ thin film photoanode"

Correspondence Authors: Prof. Yanbo Li

Authors: Yequan Xiao, Mr Zeyu Fan, Mamiko Nakabayashi, Qiaoqiao Li, Prof. Liujiang Zhou, Prof. Qian Wang, Prof. Changli Li, Prof. Naoya Shibata, Prof. Kazunari Domen

Dear Reviewers:

We once again appreciate the referee's careful review and valuable comments and suggestions. The constructive comments made by the reviewers have helped to further improve the quality of our work. We are very glad that the reviewers are satisfied with our revision and are now supportive for publication in *Nature Communications*. The point-by-point responses to the reviewers' comments are provided below:

Reviewer #1

Remarks to the Author: *The authors have adequately addressed all of the minor concerns that I had with the manuscript, including conducting additional XPS depth profile experiments. I am satisfied with their changes and not (now) recommend their work for publication.*

Our Response: We are pleased to hear that the reviewer is satisfied with our response to the reviewer's comments. We thank the reviewer again for his/her comments on further improving the overall quality of our manuscript.

Reviewer #2:

Remarks to the Author: *The authors have addressed my concerns of the original manuscript clearly and professionally. I recommend the publication of this paper in your journal at this point.*

Our Response: We are very glad that the reviewer finds our response clear and professional and thank the reviewer for recommending our work for publication. We further would like to thank the reviewer for constructive comments which helped to substantially improve the quality of the paper.

Reviewer #3:

Remarks to the Author: This manuscript has been properly revised and can be published as it is.

Our Response: We genuinely appreciate the reviewer's positive evaluation of our work and thank the reviewer for his/her valuable comments which have helped to substantially improve the overall quality of our manuscript.